# Receptor-mediated mitophagy regulates EPO production and protects against renal anemia

Guangfeng Geng[1†], Jinhua Liu[2†], Changlu Xu[2†], Yandong Pei[1], Linbo Chen[1], Chenglong Mu[1], Ding Wang[2], Jie Gao[2], Yue Li[2], Jing Liang[2], Tian Zhao[1], Chuanmei Zhang[1], Jiaxi Zhou[2], Quan Chen[1]*, Yushan Zhu[1]*, Lihong Shi[2]*

[1]State Key Laboratory of Medicinal Chemical Biology, Frontiers Science Center for Cell Responses, Tianjin Key Laboratory of Protein Science, College of Life Sciences, Nankai University, Tianjin, China; [2]State Key Laboratory of Experimental Hematology, National Clinical Research Center for Blood Diseases, Institute of Hematology & Blood Diseases Hospital, Chinese Academy of Medical Sciences & Peking Union Medical College, Tianjin, China

**Abstract** Erythropoietin (EPO) drives erythropoiesis and is secreted mainly by the kidney upon hypoxic or anemic stress. The paucity of EPO production in renal EPO-producing cells (REPs) causes renal anemia, one of the most common complications of chronic nephropathies. Although mitochondrial dysfunction is commonly observed in several renal and hematopoietic disorders, the mechanism by which mitochondrial quality control impacts renal anemia remains elusive. In this study, we showed that FUNDC1, a mitophagy receptor, plays a critical role in EPO-driven erythropoiesis induced by stresses. Mechanistically, EPO production is impaired in REPs in *Fundc1⁻ᐟ⁻* mice upon stresses, and the impairment is caused by the accumulation of damaged mitochondria, which consequently leads to the elevation of the reactive oxygen species (ROS) level and triggers inflammatory responses by up-regulating proinflammatory cytokines. These inflammatory factors promote the myofibroblastic transformation of REPs, resulting in the reduction of EPO production. We therefore provide a link between aberrant mitophagy and deficient EPO generation in renal anemia. Our results also suggest that the mitochondrial quality control safeguards REPs under stresses, which may serve as a potential therapeutic strategy for the treatment of renal anemia.

**\*For correspondence:**
chenq@ioz.ac.cn (QC);
zhuys@nankai.edu.cn (YZ);
shilihongxys@ihcams.ac.cn (LS)

[†]These authors contributed equally to this work

**Competing interests:** The authors declare that no competing interests exist.

## Introduction

Erythropoiesis is a highly dynamic and tightly regulated process in which red blood cells are generated from immature progenitors in the bone marrow (*An et al., 2015*; *Nandakumar et al., 2016*). This process is mainly driven by the cytokine glycoprotein erythropoietin (EPO) (*Kuhrt and Wojchowski, 2015*). Upon the engagement of EPO with its receptor, erythropoietin receptor (EPOR), the Janus kinase (JAK) signal transducer and activator of transcription (STAT) pathway is activated to prevent committed erythroid progenitors from apoptosis and to facilitate their proliferation and differentiation into red blood cells (*Nandakumar et al., 2016*). During the final maturation step of erythropoiesis, reticulocytes need to remove all of the remaining organelles, including mitochondria, endocytic vesicles, ribosomes, Golgi cisternae, and endoplasmic reticulum (*Koury et al., 2005*; *Gronowicz et al., 1984*). Among them, mitochondria are the most abundant organelles for removal (*Zhang et al., 2015*). Prior studies have demonstrated that defects in mitochondrial removal are linked to the dysfunction of red blood cells and can lead to anemia (*Kundu et al., 2008*; *Mortensen et al., 2010*; *Honda et al., 2014*; *Nishida et al., 2009*; *Schweers et al., 2007*;

*Sandoval et al., 2008*). Therefore, a deeper understanding of the molecular basis of mitochondrial quality control is of great significance for dissecting normal erythropoiesis and could elucidate the pathogenesis of dyserythropoietic diseases.

As one of the key regulators of erythropoiesis, EPO is primarily secreted by the kidneys during adulthood by the renal EPO-producing cells (REPs) located in the renal cortex and outer medulla (*Pan et al., 2011*; *Yamazaki et al., 2013*). REPs possess functional plasticity for EPO production in response to hypoxia and other microenvironmental changes (*Souma et al., 2015*). For instance, hypoxic stresses stabilize hypoxia-inducible factors (HIFs), especially HIF2α, which binds to the enhancer region of HIF binding element of the *EPO* gene and upregulates its expression (*Kuhrt and Wojchowski, 2015*). Upon inflammation or with chronic diseases, however, proinflammatory cytokines, such as TNFa and IL1b, promote the myofibroblastic transition of REPs (*Asada et al., 2011*), concomitantly repress EPO production in REPs (*Frede et al., 1997*), and ultimately lead to anemia (*Weiss and Goodnough, 2005*; *Guo et al., 1999*; *Inoue et al., 2010*). Although mitochondrial quality control exerts protective roles in a variety of renal disorders (*Liu et al., 2014*), whether and how it might be implicated in REPs during chronic or inflammatory diseases remain unclear.

Mitochondria are essential organelles that maintain cellular energy and redox homeostasis and are the major source and target of intracellular oxidative stress (*Wei et al., 2015*). They also play central roles in regulating innate immunity and apoptosis (*Wei et al., 2015*). Maintaining normal mitochondrial activities and homeostasis is of great importance for cellular and organismal metabolisms and functions (*Simon et al., 2000*; *Yakes and Van Houten, 1997*). Mitophagy, a selective type of autophagy that allows the elimination of unwanted or damaged mitochondria, is the major mechanism sustaining mitochondrial homeostasis and quality control. Mitophagy can be mediated by the PTEN-induced putative kinase 1-Parkin (PINK1-PAKN) pathway and/or by mitophagic receptors residing in the outer mitochondrial membrane (*Wei et al., 2015*; *Wu and Chen, 2015*), such as BCL2 Interacting Protein 3 Like (BNIP3L, also named as NIX) (*Novak et al., 2010*), BCL2 interacting protein 3 (BNIP3) (*Hanna et al., 2012*), FUN14 domain containing 1 (FUNDC1) (*Liu et al., 2012a*; *Chen et al., 2014*), Prohibitin2 (PHB2) (*Wei et al., 2017*), and FKBP Prolyl Isomerase 8 (FKBP8) (*Saita et al., 2013*) in mammalian cells. It has been demonstrated that receptor-mediated mitophagy is important for erythroid maturation (*Kundu et al., 2008*; *Mortensen et al., 2010*; *Honda et al., 2014*; *Nishida et al., 2009*; *Schweers et al., 2007*; *Sandoval et al., 2008*). For example, *Bnip3l*$^{-/-}$ mice exhibit anemia and reticulocytosis with defective mitophagy due to the persistence of the mitochondrial membrane potential and the resultant depolarization in the mitochondria (*Schweers et al., 2007*; *Sandoval et al., 2008*). However, in the reticulocytes of the *Bnip3l*$^{-/-}$ mice, mitochondrial removal still occurs, raising the possibility that additional factors are also required for mitochondrial degradation during red blood cell maturation.

We have previously shown that FUNDC1 functions as a specific mitophagy receptor in response to mitochondrial stresses, such as hypoxia (*Liu et al., 2012a*; *Chen et al., 2017*). FUNDC1 regulates mitochondrial homeostasis, including mitochondrial fission and fusion, content alteration, quality surveillance and physical interactions with other organelles (*Li et al., 2019*). Deletion of *Fundc1* results in the accumulation of dysfunctional mitochondria and the activation of inflammasomes and is associated with a wide range of pathological conditions, such as hepatocarcinogenesis (*Li et al., 2018*) and heart failure (*Wu et al., 2017*). Although the role of FUNDC1 has been studied in an array of cellular processes, whether it functions during EPO-driven erythropoiesis is still unexplored. In this study, using *Fundc1* germline deletion mice (*Fundc1*$^{-/-}$ mice), we found that despite being dispensable for erythroid maturation, FUNDC1 regulates EPO production during stress erythropoiesis. Mechanistically, enhanced inflammatory responses in REPs lead to insufficient EPO production as a consequence of compromised mitophagy. Together, our studies reveal a link between deficient mitophagy and reduction in EPO production and demonstrate that mitochondrial quality control safeguards REPs under stresses.

## Results

### *Fundc1* is implicated in stress but not steady-state erythropoiesis

Previous studies have shown that BNIP3L, an important mitophagy receptor, is involved in mitochondrial elimination in reticulocytes and proper functions in red blood cells (*Kundu et al., 2008*;

*Mortensen et al., 2010*; *Honda et al., 2014*; *Nishida et al., 2009*; *Sandoval et al., 2008*). These findings led us to dissect the function of FUNDC1, another key mitophagy receptor we previously characterized (*Liu et al., 2012a*), in erythroid development and differentiation. We found that in *Fundc1⁻/⁻* mice, the hemogram parameters of the peripheral blood, including erythroid parameters such as red blood cell (RBC) count, hemoglobin amount, hematocrit (HCT, measuring the volume percentage of red blood cells in whole blood), and reticulocytes, were comparable to those in the corresponding wild-type (WT) littermates at 8–10 weeks old (*Figure 1—figure supplement 1A* and *Supplementary file 1*). Despite evident mitochondrial retention in erythroid cells of *Bnip3l⁻/⁻* mice (*Sandoval et al., 2008*), negligible mitochondrial accumulation was detected in the *Fundc1⁻/⁻* cells collected from the peripheral blood, spleen and bone marrow (*Figure 1—figure supplement 1B–D*). In addition, *Fundc1* deletion did not affect erythroid lineage commitment and differentiation (*Figure 1—figure supplement 1E*), generation of hematopoietic stem and progenitors (*Figure 1—figure supplement 1F*), or differentiation of other non-erythroid, hematopoietic lineages (*Figure 1—figure supplement 1G,H*) in peripheral blood, spleen, and bone marrow, respectively. Together, these observations suggest that FUNDC1 is dispensable for steady-state erythropoiesis and possibly for hematopoiesis.

The role of FUNDC1 in mitophagy induced by hypoxia and other stresses led us to speculate that FUNDC1 might play a role in stress erythropoiesis. We treated the *Fundc1⁻/⁻* mice (*Zhang et al., 2016*) with phenylhydrazine (PHZ, 100 mg/kg), a common agent to induce acute hemolytic anemia via the induction of lysis of mature red blood cells, and then tracked hemogram in peripheral blood for a week (*Figure 1A*). We found that when compared with WT counterparts, *Fundc1⁻/⁻* mice exhibited low survival with PHZ treatment (p = 0.04, n = 30) (*Figure 1B*). Time-course hemogram examination showed reduced RBC numbers, lower HGB levels, and decreased HCT values in PHZ-*Fundc1⁻/⁻* mice than those of controls (*Figure 1C*, *Figure 1—figure supplement 2A* and *Supplementary file 1*).

Given the established function of *Fundc1* in mitophagy, we next asked whether reduced RBCs in PHZ-*Fundc1⁻/⁻* mice might be caused by the defects in mitophagy, which may block the maturation of reticulocytes to red blood cells. Given that the reticulocytosis occurred in WT and *Fundc1⁻/⁻* mice upon PHZ treatment (*Figure 1—figure supplement 2A*), we then FACS-sorted Ter119⁺CD71⁺ reticulocytes in the peripheral blood of WT and *Fundc1⁻/⁻* mice on day 5 after PHZ treatment (*Figure 1—figure supplement 2B*). The isolated reticulocytes were morphologically confirmed by Giemsa staining (*Figure 1—figure supplement 2C*, left) and brilliant cresyl blue staining (*Figure 1—figure supplement 2C*, right). Next, the reticulocytes were cultured to maturation *in vitro* for 3 days using a previously described method (*Koury et al., 2005*). The brilliant cresyl blue staining showed progressive maturation of reticulocytes in WT mice after 3 days of culture (*Figure 1—figure supplement 2C*). Unexpectedly, there was no detectable difference in the maturation of reticulocytes between WT and *Fundc1⁻/⁻* mice (*Figure 1—figure supplement 2C*). Consistent with these findings, the mitochondrial mass, as measured with MitoTracker Green, exhibited little difference between WT and *Fundc1⁻/⁻* mice during reticulocyte maturation (*Figure 1—figure supplement 2D*). Thus, *Fundc1*-mediated mitophagy seems dispensable for reticulocyte maturation.

To further decipher the cause of reduced RBCs, we assessed erythropoiesis in the spleen of PHZ-*Fundc1⁻/⁻* mice because stress erythropoiesis primarily occurs in the mouse spleen. We found that the total number of erythroid cells in the spleen of PHZ-*Fundc1⁻/⁻* mice was reduced compared to WT mice after 48 hr of treatment (p<0.05, *Figure 1D*), while there were no apparent differences in spleen size, weight, and color (*Figure 1—figure supplement 2E*). Subsequent analysis revealed a transient reduction of the erythroid compartment such as the R2 population (CD71^hiTer119⁺) (mean 8% vs 2%, p<0.01) (*Figure 1E,F* and *Figure 1—figure supplement 2F*). Nonetheless, the decreased R2 population was unlikely to be due to elevated apoptosis (*Figure 1—figure supplement 2G*), accumulation of mitochondrial mass (*Figure 1G*), aberrant mitochondrial membrane potential (*Figure 1H*), or enhanced mitochondrial reactive oxygen species (ROS) level (*Figure 1I*). Additionally, we examined splenic progenitors including CMP, MEP, and GMP in PHZ-*Fundc1⁻/⁻* mice and found no obvious alterations in either the percentage or the total cell number (*Figure 1—figure supplement 2H,I*).

Together, our results suggest that FUNDC1-mediated mitophagy is dispensable for steady-state erythropoiesis, whereas it is involved in PHZ-induced stress erythropoiesis.

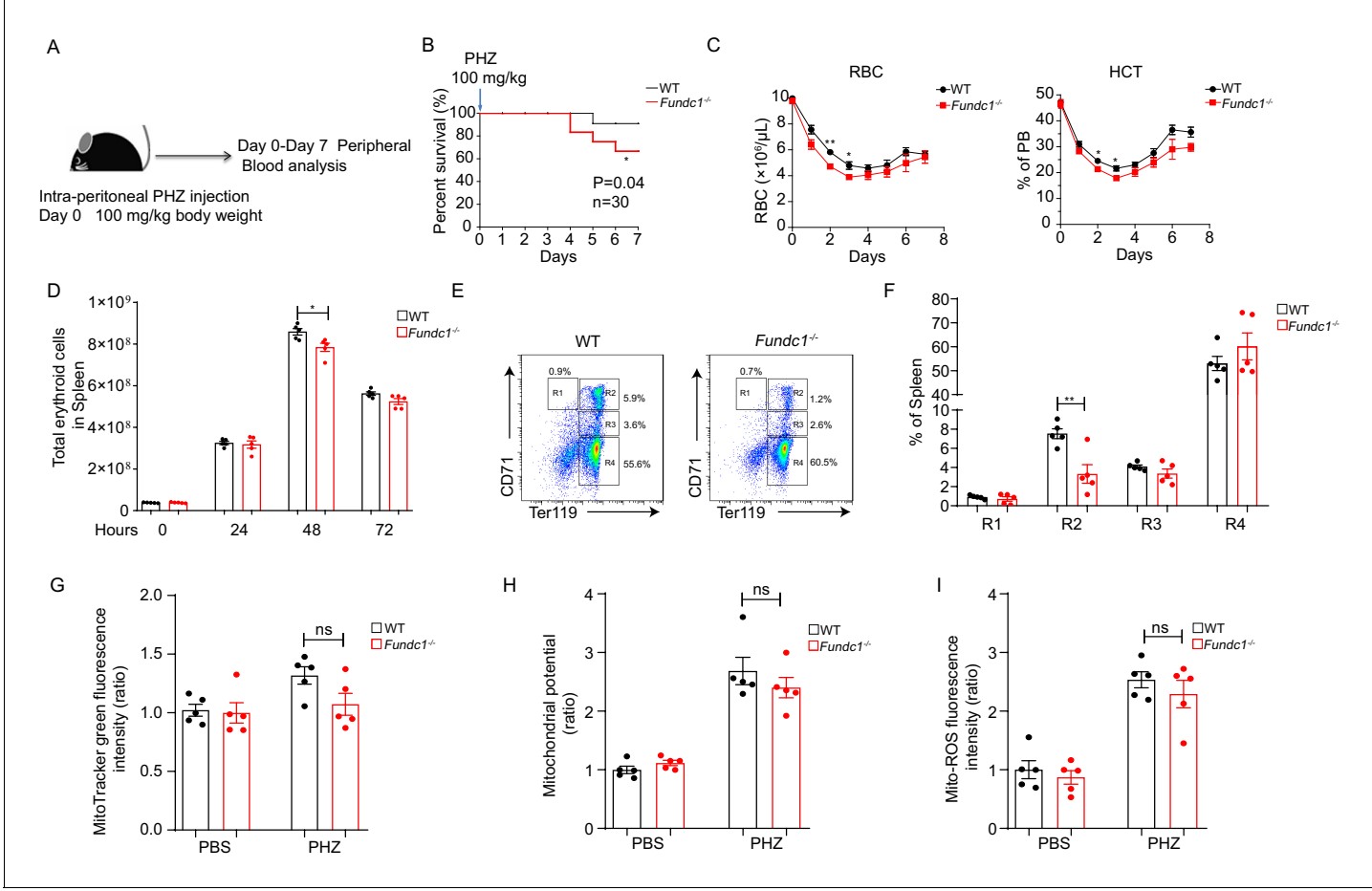

**Figure 1.** Defects of splenic erythroid progenitor production in *Fundc1*⁻/⁻ mice during stress erythropoiesis. (**A**) Schematic diagram of PHZ-induced stress erythropoiesis. Mice around 8–10 weeks old were treated with a single dose of PHZ (100 mg/kg body weight) to induce acute anemia, and multiple examinations were taken in the following week. (**B**) Survival curves of PHZ- treated WT control littermates and *Fundc1*⁻/⁻ mice (n = 30 for each group). (**C**) Hemogram parameters of red blood cell (RBC) counts and hematocrit (HCT) in the peripheral blood of WT controls and *Fundc1*⁻/⁻ mice from day 0 to 7 after PHZ treatment. (**D**) Total erythroid cell number in spleen of WT and *Fundc1*⁻/⁻ mice at 0, 24, 48, and 72 hr after PHZ treatment, respectively. (**E, F**) Flow cytometry analysis of erythroid differentiation with using CD71 and Ter119 from the spleen of WT and *Fundc1*⁻/⁻ mice after 48 hr of PHZ treatment. The representative flow cytometry diagram (**E**) and the corresponding quantitative analysis (**F**) are shown. (**G–I**) The bar graphs showing the normalized intensity of fluorescence by using flow cytometry of mitochondrial mass assayed by MitoTracker Green (**G**), membrane potential by TMRM (**H**) and ROS level by MitoSOX (**I**) within splenic R2 erythroid cells of WT or *Fundc1*⁻/⁻ mice after 48 hr of PHZ treatment, respectively. PBS acts as the control group for PHZ treatment. For each experiment (**C–I**), n = 3–5 mice for each group. Individual mice are represented by symbols. Data shown are representative of at least three independent experiments. Similar results were found in each experiment. All data are mean ± SEM; *p<0.05, **p<0.01. Statistical significance was analyzed by using the two-tailed unpaired Student's *t*-test.

The online version of this article includes the following source data and figure supplement(s) for figure 1:

**Source data 1.** Erythroid progenitor production during stress erythropoiesis.
**Figure supplement 1.** Steady-state hematopoiesis and erythropoiesis in *Fundc1*⁻/⁻ mice.
**Figure supplement 1—source data 1.** List of steady-state hematopoiesis and erythropoiesis in WT, *Fundc1*⁻/⁻ and *Bnip3l*⁻/⁻ mice.
**Figure supplement 2.** FUNDC1-mediated mitophagy is dispensable for reticulocyte maturation under stress erythropoiesis.
**Figure supplement 2—source data 1.** Reticulocyte maturation of WT and *Fundc1*⁻/⁻ mice.

## Hypoactivation of JAK-STAT signaling underlies the reduction of erythroid population

To uncover the molecular mechanism governing the reduction of R2 population, we next conducted high-throughput transcriptome profiling using FACS-purified R2 cells from the spleen of WT and *Fundc1*⁻/⁻ mice after 48 hr of PHZ treatment. We identified 1065 significant differentially expressed genes (DEGs) in total (*Figure 2A*). Kyoto Encyclopedia of Genes and Genomes (KEGG) enrichment

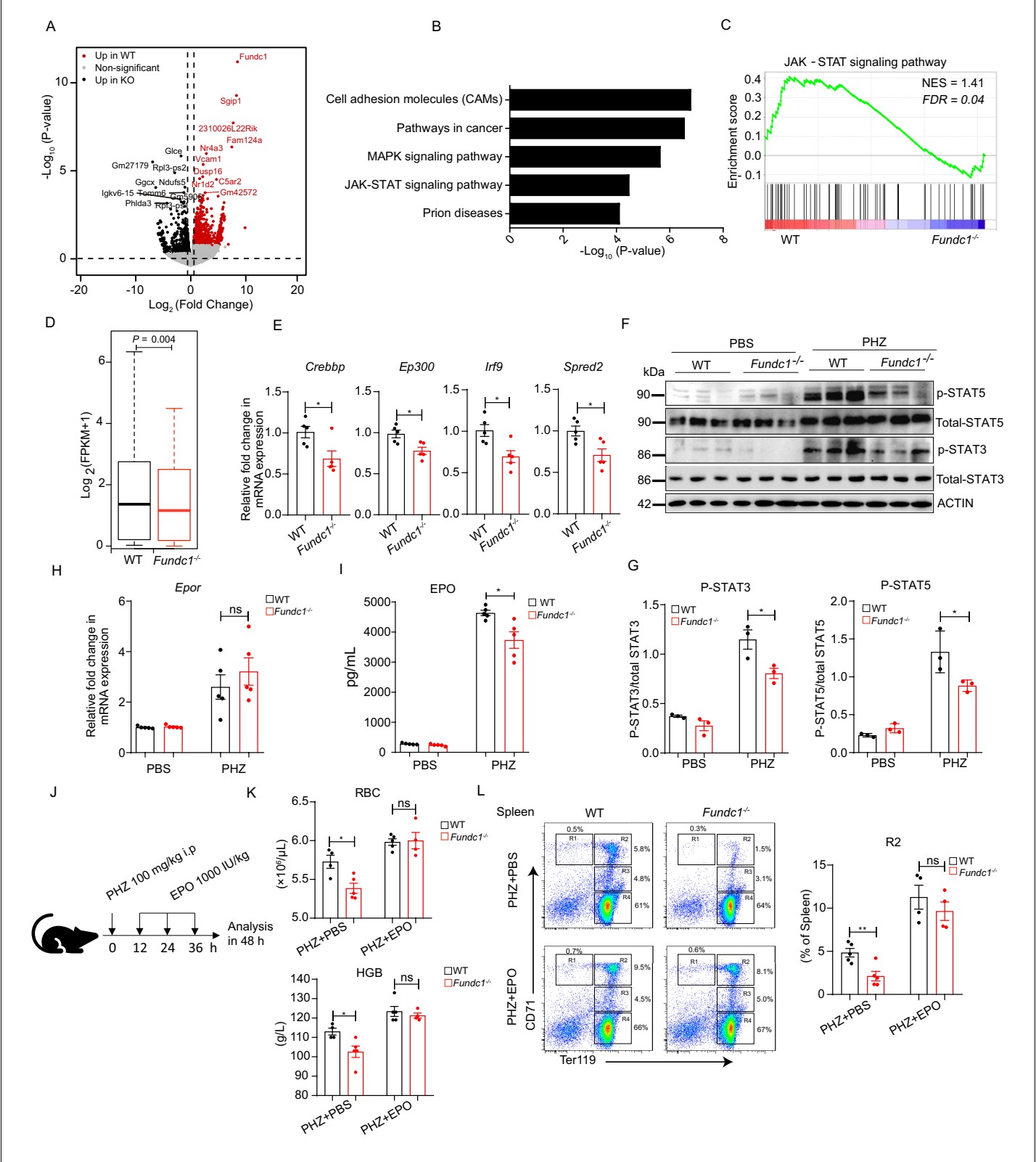

**Figure 2.** Hypoactivation of JAK-STAT signaling underlies the comprised erythroid progenitor production in *Fundc1⁻/⁻* mice during stress erythropoiesis. (**A**) Volcano Plot illustrates the differentially expressed genes (DEGs) of FACS-purified erythroid progenitors (R2 populations) in WT (n = 3) and *Fundc1⁻/⁻* (n = 3) spleen at 48 hr after PHZ treatment. Top 10 genes in each group were labeled. (**B**) Top 5 KEGG functional enrichment pathways analysis of downregulated DEGs in *Fundc1⁻/⁻* R2 cells. (**C**) GSEA showing the enrichment of the JAK-STAT signaling pathway in splenic R2 cells from WT and

*Figure 2 continued on next page*

*Figure 2 continued*

*Fundc1*$^{-/-}$ mice. Normalized enrichment score (NES) and false discovery rate (FDR) are shown. (D) The box plot depicts the quantitative analysis of the genes in JAK-STAT signaling pathway in splenic R2 cells of WT and *Fundc1*$^{-/-}$ mice. The boxes represent the median and quartile of the sum of log$_2$ transformed (FPKM +1). p-value was determined by using paired Wilcoxon signed-rank Test. (E) The expression of key components in the JAK-STAT pathway in splenic R2 cells from WT and *Fundc1*$^{-/-}$ mice. (F) Western blotting of STAT3, STAT5 and phosphorylated STAT3 and STAT5 in FACS-sorted R2 populations in spleen of WT and *Fundc1*$^{-/-}$ mice after 48 hr of PBS or PHZ treatment, respectively. ACTIN was used as a loading control. n = 3. (G) The quantification of the signal intensity of phosphorylated STAT3 and STAT5 out of the total STAT3 and STAT5 in (F). (H) The expression of *Epor* mRNA in the spleen of WT and *Fundc1*$^{-/-}$ mice after 48 hr of PBS or PHZ treatment. (I) The serum EPO concentration in *Fundc1*$^{-/-}$ and WT mice after 48 hr of PBS or PHZ treatment. (J) Schematic illustrating the administration of EPO to PHZ-*Fundc1*$^{-/-}$ and PHZ-WT mice. (K) RBC number (upper) and HGB level (bottom) after EPO supplement in PHZ-*Fundc1*$^{-/-}$ and PHZ-WT mice. (L) Erythroid differentiation in spleen of PHZ-*Fundc1*$^{-/-}$ and PHZ-WT after EPO administration. The left panel shows representative FACS plots and right bar graph shows the statistical quantification of the percentage of the R2 compartment. For each experiment (E–L), n = 3–5 mice for each group. Individual mice are represented by symbols. Data shown are representative of at least three independent experiments. Similar results were found in each experiment. All data are mean ± SEM; *: p<0.05; ns: no statistical significance. Statistical significance was analyzed by using the two-tailed unpaired Student's *t*-test unless stated otherwise.

The online version of this article includes the following source data and figure supplement(s) for figure 2:

**Source data 1.** The JAK-STAT signaling in WT and *Fundc1*$^{-/-}$ mice during stress erythropoiesis.

**Figure supplement 1.** *Fundc1*$^{-/-}$ mice resembles WT mice upon phlebotomy.

**Figure supplement 1—source data 1.** Erythropoiesis in WT and *Fundc1*$^{-/-}$ mice upon phlebotomy.

analysis with DEGs revealed differential regulation of key signaling pathways and biological processes (*Figure 2B*). We found that the JAK-STAT signaling pathway was among the top five significantly enriched pathways (*Figure 2B*), in agreement with gene set enrichment analysis (GSEA) (*Figure 2C*) and gene set quantitative analysis (*Figure 2D*). The key components and downstream targets of JAK-STAT signaling pathway, such as *Crebbp*, *Ep300*, *Irf9*, and *Spred2*, were also downregulated in *Fundc1*$^{-/-}$ R2 cells (*Figure 2E*).

To further confirm the hypoactivation of JAK-STAT signaling, we analyzed the total and phosphorylated levels of STAT5 and STAT3 proteins in FACS-purified R2 cells, which are central players in the JAK-STAT pathway during erythropoiesis (*Richmond et al., 2005*). In WT R2 cells, PHZ treatment had limited effects on the total levels of STAT5 and STAT3 (*Figure 2F*). However, the levels of phosphorylated STAT5 and STAT3 were markedly elevated (*Figure 2F,G*), indicating that the JAK-STAT signaling pathway in R2 cells was activated upon PHZ-induced hemolytic stress. In striking contrast, the levels of phosphorylation of STAT5 and STAT3 were significantly lower in PHZ-*Fundc1*$^{-/-}$ R2 cells than those in the WT counterparts (*Figure 2F,G*). Thus, the JAK-STAT signaling pathway is hypoactivated in PHZ-treated *Fundc1*$^{-/-}$ R2 cells.

During erythropoiesis, the JAK-STAT signaling pathway acts downstream of the EPO receptor (EPOR) and is regulated by the serum EPO level (*Kuhrt and Wojchowski, 2015*). EPOR, a transmembrane glycoprotein, is expressed in erythroid progenitors (colony-forming unit-erythroid: CFU-E), proerythroblasts, and early basophilic erythroblasts (*Kuhrt and Wojchowski, 2015*). These erythroid progenitors and precursors are the major components of the R2 population (*Socolovsky et al., 2001*), which may account for the sensitivity of the R2 compartment to EPO paucity. We then investigated the *Epor* expression in R2 cells in WT and *Fundc1*$^{-/-}$ mice with or without PHZ treatment and found no apparent changes when *Fundc1* was deleted (*Figure 2H*). However, enzyme-linked immunosorbent assay (ELISA) experiments showed that the serum EPO level was significantly reduced in PHZ-*Fundc1*$^{-/-}$ mice compared with the corresponding controls (6,100 vs 4,400 and 4,500 vs 3,500 pg/mL after 24 and 48 hr of PHZ treatment, respectively, p<0.05) (*Figure 2—figure supplement 1A* and *Figure 2I*). In contrast, a large number of inflammatory cytokines in the serum, such as IL6 and IL1, exhibited a modest increase in PHZ-*Fundc1*$^{-/-}$ mice (*Figure 2—figure supplement 1B*). More importantly, exogenous administration of EPO (1,000 IU/kg) to PHZ-*Fundc1*$^{-/-}$ mice (*Figure 2J*) showed the comparable peripheral RBC number and HGB level with their WT counterparts (*Figure 2K*). Erythroid differentiation in the spleen of PHZ-*Fundc1*$^{-/-}$ mice was restored to the levels of PHZ-WT controls (particularly of R2) as well (*Figure 2L*). These results suggested that insufficient EPO supply underlies the reduced RBC number in PHZ-*Fundc1*$^{-/-}$ mice. In contrast, no difference in EPO production was observed between *Fundc1*$^{-/-}$ mice and the control mice in the pure anemic model as determined via phlebotomy of 500 μL blood for three consecutive days (*Figure 2—figure*

*supplement 1C–G*), suggesting that the reduced EPO production in PHZ-*Fundc1*[-/-] mice most likely arises from the oxidative stress of PHZ.

Taken together, these results suggest the following scenario: *Fundc1* deletion reduces EPO secretion and consequently leads to hypoactivation of JAK-STAT signaling in erythroid R2 cells in the spleen during stress erythropoiesis, which ultimately results in the reduction of the red blood cell count in *Fundc1*[-/-] mice with PHZ treatment.

## *Fundc1* deletion causes renal injury

The kidney is the predominant organ for EPO production in adults. EPO is produced by REPs that are responsible for approximately 90% of total EPO synthesis (*Nolan and Wenger, 2018*). We therefore examined the impact of PHZ on the kidney in *Fundc1*[-/-] mice after 48 hr of treatment. We found that *Fundc1* was highly and ubiquitously expressed in the kidney (*Figure 3—figure supplement 1A, B*). In addition, the mRNA level of *Epo* in kidney of *Fundc1*[-/-] mice was indeed significantly lower than that of its WT counterparts at 48 hr after PHZ treatment (130 vs 61, p<0.05) (*Figure 3A*). We also detected the elevated expression of two kidney injury markers, Lipocalin-2 (*Lcn2*) (*Moschen et al., 2017*) and kidney injury molecule-1 (*Kim1*) (*Humphreys et al., 2013*; *Figure 3B*), although hematoxylin and eosin (H&E) staining illustrated no major structural alterations in the kidneys of PHZ-*Fundc1*[-/-] mice (*Figure 3—figure supplement 1C*).

It has been shown previously that EPO production is almost exclusively controlled at the transcription level by HIFs, predominantly by HIF2α (*Kapitsinou et al., 2010*; *Rankin et al., 2007*; *Gruber et al., 2007*). These findings led us to examine the potential link between EPO and HIFs in PHZ-treated *Fundc1*[-/-] mice. To this end, we first measured the expression of HIF2α and its downstream targets, *Vegf* and *Slc2a1*, in the kidneys of PHZ-treated *Fundc1*[-/-] mice. Although HIF2α protein levels were sharply enhanced in kidneys of both WT and *Fundc1*[-/-] mice after PHZ treatment, the levels between WT and *Fundc1*[-/-] kidneys were similar (*Figure 3C*). Similar results were observed for the HIF2α targets *Vegf* and *Slc2a1* (*Figure 3—figure supplement 1D*). Thus, the reduction of EPO secretion is unlikely to be caused by the aberrant transcriptional regulation of *Epo*. We also excluded the possibility of apoptosis in the kidneys of PHZ-*Fundc1*[-/-] mice (*Figure 3—figure supplement 1E*).

How is EPO supply attenuated in PHZ-*Fundc1*[-/-] mice? The key role of FUNDC1 in mitophagy led us to further test the efficiency of mitophagy. In immunohistochemical analysis of mitochondrial membrane protein Prohibitin 2 (PHB2), we found that PHZ treatment generally reduced the accumulation of total mitochondrial mass (*Figure 3D*), suggesting that PHZ stress could promote the mitophagic process. In contrast, *Fundc1* deletion resulted in more retainment of mitochondria in the kidneys than in the WT controls after PHZ treatment (*Figure 3D*), indicating that mitophagy was attenuated in *Fundc1*[-/-] kidneys under PHZ-induced stresses. To further support this notion, we found that the mitochondrial membrane proteins TIMM23, COXIV, and TOMM20 were more abundant in PHZ-*Fundc1*[-/-] renal cells than in the control cells (*Figure 3E,F*).

The excessive mitochondrial accumulation may be due to either enhanced mitochondrial biogenesis or inefficient mitochondrial elimination. To distinguish between these two possibilities, we next examined mitochondrial biogenesis by determining the expression of the mitochondrial constitutive components of ATP synthase, including F1 subunit delta (*Atp5d*), nuclear respiratory factor 1 (*Nrf1*), and transcription factor A mitochondria (*Tfam*), and found no apparent differences between the kidneys of the PHZ-*Fundc1*[-/-] mice and the PHZ-treated control mice (*Figure 3G*). Thus, *Fundc1* deletion has little impact on mitochondrial biogenesis. Instead, the expression of autophagic proteins of LC3-II and P62 in PHZ-*Fundc1*[-/-] renal cells was reduced (*Figure 3E,F*), suggesting that the accumulation of mitochondrial content is caused by defective autophagy.

The mitochondria, when damaged, are retained in the cytoplasm and inclined to release their DNA (mtDNA). Thus, to further confirm that deficient mitophagy was the cause of the accumulation of damaged mitochondria in PHZ-*Fundc1*[-/-] mouse kidneys, we examined the level of total and leaked cytosolic mtDNA from the lysed kidneys in PHZ-*Fundc1*[-/-] mice. With this aim, we removed the nuclei (to exclude the genomic DNA contamination) and mitochondria (to exclude intrinsic mitochondrial DNA contamination) (see Materials and methods) and discovered that the total level of mtDNA was elevated, while the cytosolic mtDNA was indeed induced in PHZ-*Fundc1*[-/-] renal cells (*Figure 3H,I*). Because cytosolic mtDNA acts as a critical trigger for cellular inflammatory responses (*Allison, 2019*), we then determined the level of inflammatory markers in renal cells and observed

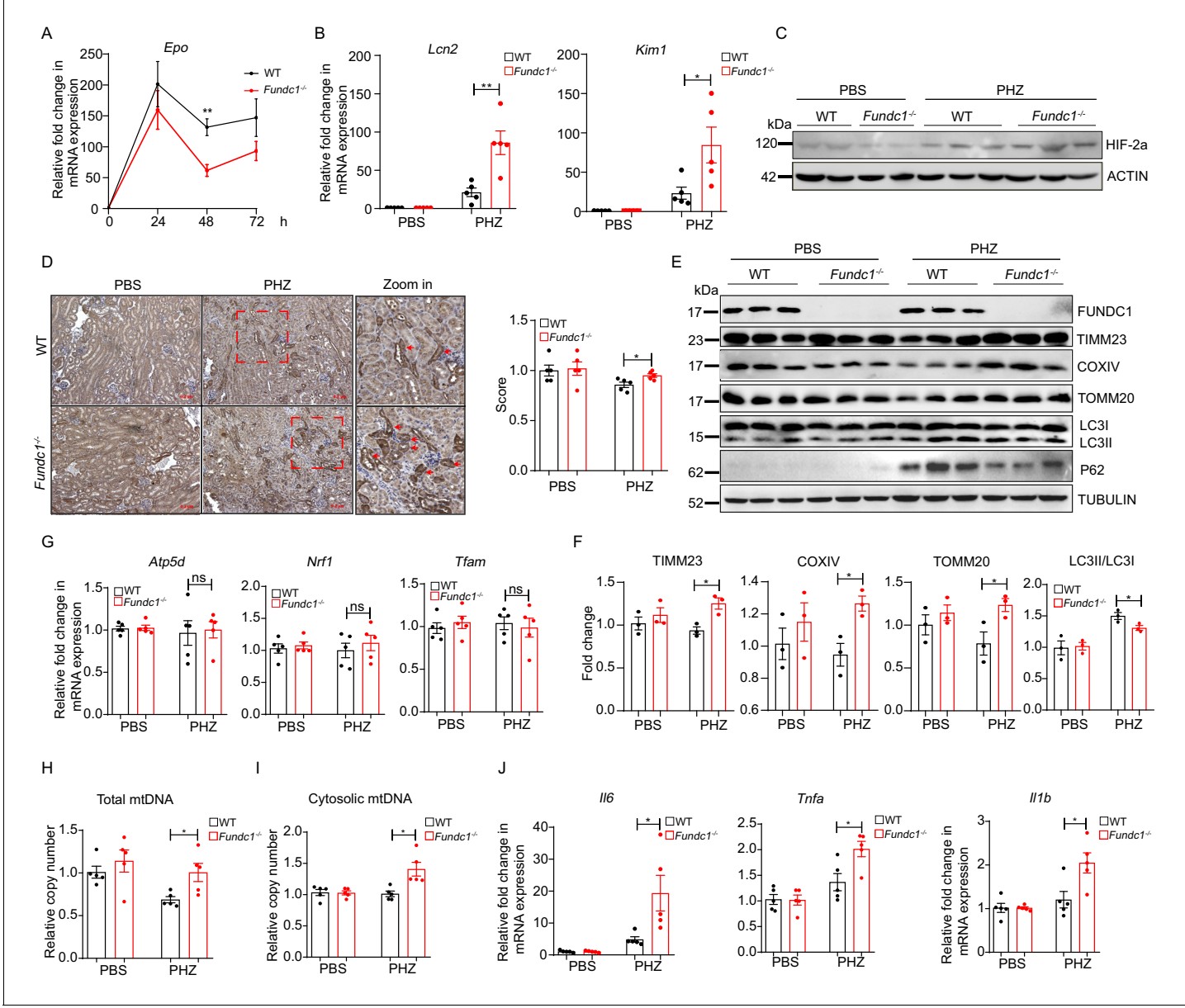

**Figure 3.** MtDNA cytosolic translocation caused by insufficient mitophagy induces elevated inflammation and severer renal injury in *Fundc1*[-/-] mice upon PHZ treatment. (**A**) qRT-PCR analysis showing the kinetics of *Epo* mRNA expression in kidneys of WT and *Fundc1*[-/-] mice after PBS or PHZ treatment. (**B**) Relative mRNA levels of kidney injury marker *Lcn2* and *Kim1* in kidneys of WT and *Fundc1*[-/-] mice with or without PHZ treatment. (**C**) Western blotting showing the HIF2α expression in whole kidney lysate in WT and *Fundc1*[-/-] mice with or without PHZ treatment. ACTIN was used as a loading control. (**D**) Immunohistochemistry of mitochondrial protein PHB2 in kidneys of WT and *Fundc1*[-/-] mice with or without PHZ treatment, respectively. The indicated area (blue dash box) is zoomed in and intense signals of PHB2 are marked with red arrows. The right bar graph presents the quantitative assay measured by ImageJ. Scale bars, 200 mm; original magnification, 10×. (**E, F**) Western blotting (**E**) and corresponding quantification (**F**) showing the protein levels of mitochondrial membrane proteins (TIMM23, COXIV, and TOMM20) and autophagic proteins (P62, LC3II/I, and their ratio) in kidneys of WT and *Fundc1*[-/-] mice with or without PHZ treatment. TUBULIN was used as a loading control. (**G**) Relative mRNA levels of *Atp5d*, *Nrf1, and Tfam* in kidneys of WT and *Fundc1*[-/-] mice with or without PHZ treatment. (**H, I**) qRT-PCR shows the total and cytosolic mtDNA in kidneys of WT and *Fundc1*[-/-] mice with or without PHZ treatment. (**J**) The mRNA levels of *Il1b*, *Il6*, and *Tnfa* in kidneys treated with PBS or PHZ. For each experiment, n = 3–5 mice for each group. Individual mice are represented by symbols. Data shown are representative of at least three independent experiments. Similar results were found in each experiment. All data are mean ± SEM; *p<0.05, **p<0.01, ns: no statistical significance. Statistical significance was analyzed by using the two-tailed unpaired Student's *t*-test.

The online version of this article includes the following source data and figure supplement(s) for figure 3:

**Source data 1.** MtDNA cytosolic translocation and severer renal injury in *Fundc1*[-/-] mice upon PHZ treatment.
**Figure supplement 1.** The histology of kidney of *Fundc1*[-/-] mice after PHZ treatment.

*Figure 3 continued on next page*

*Figure 3 continued*

**Figure supplement 1—source data 1.** The histology of kidney of *Fundc1*[-/-] mice after PHZ treatment.

significant upregulation of proinflammatory cytokines, including *Il6*, *Tnfa*, and *Il1b* in PHZ-treated kidneys of *Fundc1*[-/-] mice (*Figure 3J*). Thus, *Fundc1* deletion causes renal injury upon PHZ treatment by activating inflammation as a result of defective mitophagy.

## Deficient mitophagy leads to reduced EPO production and increased inflammation

The inflammatory milieu in the kidney is one of the major contributors to renal fibrosis (*Lv et al., 2018*), which in turn facilitates REP conversion into myofibroblasts and the concomitant loss of EPO production capability (*Asada et al., 2011*; *Souma et al., 2013*). Therefore, it is reasonable to propose that the low EPO yield in REPs might result from their fibrotic transformation in PHZ-*Fundc1*[-/-] mice, which is induced by activation of the inflammatory milieu in the cytosol. To test this hypothesis, we co-stained α-smooth muscle actin (α-SMA), a broadly used myofibroblast marker (*Duffield, 2014*), and platelet-derived growth factor receptor beta (PDGFRβ), a REP marker (*Asada et al., 2011*; *Gerl et al., 2016*), in the renal tissue of PHZ-treated mice. We found that REPs exhibited stronger fibrotic signatures in PHZ-*Fundc1*[-/-] mice than in the PHZ-WT controls (*Figure 4A*).

We next sought to dissect the molecular mechanism underlying the fibrotic transformation of REPs. We sorted PDGFRβ[+] REPs from WT and *Fundc1*[-/-] mice (*Asada et al., 2011*; *Lemos et al., 2018*; *Figure 4B,C*) and then examined *Epo* mRNA levels in PDGFRβ[+] and PDGFRβ[-] cells by using quantitative real-time PCR (qRT-PCR) analysis. As expected, PDGFRβ[+] cells exhibited more *Epo* production (*Figure 4D*). To elucidate the potential role of FUNDC1-mediated mitophagy in REPs, we cultured purified PDGFRβ[+] cells from WT and *Fundc1*[-/-] mice *in vitro* for 24 hr upon hypoxic stress stimulation (1% $O_2$). We found that in WT REPs, the levels of mitochondrial membrane proteins (TIMM23, COXIV, and TOMM20) were sharply reduced upon hypoxic stress, concomitant with the enhanced conversion from LC3 I to II, a hallmark of autophagy (*Kabeya et al., 2000*), indicating that hypoxia promotes the degradation of mitochondria by enhancing mitophagy (*Figure 4E,F*). In contrast, these mitochondrial membrane proteins persisted at high levels in REPs of *Fundc1*[-/-] mice under hypoxic conditions when compared with the WT counterparts (*Figure 4E,F*), suggesting that mitophagy is impaired in REPs of *Fundc1*[-/-] mice under hypoxic stress.

To unravel the mitophagic activity in PDGFRβ[+] REPs of PHZ-treated *Fundc1*[-/-] mice, we cultured purified PDGFRβ[+] cells from WT and *Fundc1*[-/-] mice with or without PHZ treatment, respectively. Similar to hypoxia, PHZ also induced the accumulation of mitochondrial mass as revealed by Mito-Tracker staining in *Fundc1*-deleted REPs (*Figure 4G*). Furthermore, to quantify the efficiency of mitophagy more directly, we crossed *Fundc1*[-/-] mice with the mitophagy reporter (mito-Keima) (*Sun et al., 2015*), a mitochondrial-targeted transgenic mice, and generated the mito-Keima/*Fundc1*[-/-] mice (*Figure 4H*). Because Keima, a coral-derived fluorescent protein, exhibits both pH-dependent excitation and resistance to lysosome proteinases, the mito-Keima mice allow for *in vivo* assessment of mitophagy based on the state of mitochondria. In mito-Keima mice, the free mitochondria (pH ~8.0) without undergoing mitophagy are green, while damaged or superfluous mitochondria are encapsulated in the lysosomes (pH ~4.5) through mitophagy, and with the altered pH value (from pH 8.0 to 4.5) the mitochondria become red. In a single cell, both green and red colors can be seen if substantial mitochondrial biosynthesis and mitophagy occur. Thus, the ratio of red and green fluorescent intensity reflects the efficiency of mitophagy *in vivo*.

We next used flow cytometry to examine PDGFRβ[+] REPs isolated from the mito-Keima/*Fundc1*[-/-] and mito-Keima/WT mice treated with either PBS or PHZ, respectively. Although the fraction of mitochondria in the lysosome was elevated in PHZ-REPs from both mito-Keima/*Fundc1*[-/-] (34.1 vs 48.4%) and mito-Keima/WT (32.7 vs 35.8%) mice, the proportion of encapsulated mitochondria was lower in PHZ-mito-Keima/*Fundc1*[-/-] than in PHZ-mito-Keima/WT mice (48.4 vs 35.8%) (*Figure 4H*). Thus, *Fundc1* deletion in REPs impairs mitophagy and results in the retention of a large number of damaged mitochondria under PHZ-induced stresses.

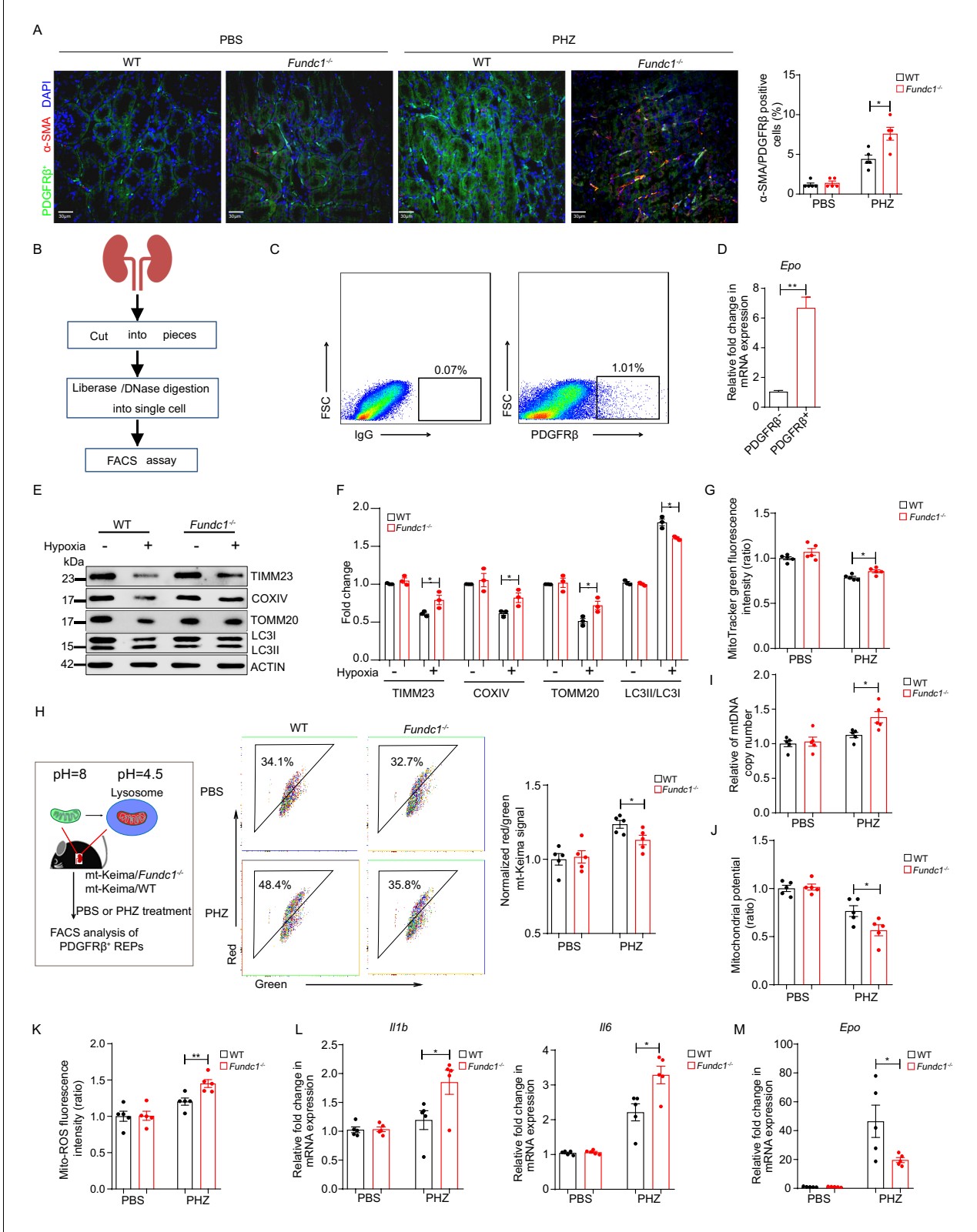

**Figure 4.** Impaired mitophagy triggers inflammation and reduced *Epo* expression in REPs. (**A**) Representative immunofluorescence images (left) of PDGFRβ and α-SMA co-staining in renal tissue sections from WT and *Fundc1*⁻/⁻ mice after 48 hr of PBS or PHZ treatment, respectively. The right bar graph shows the percentage of α-SMA positive fibrotic cells out of PDGFRβ labeled REPs. For quantification, approximately 10 fields for each mouse (200×) were randomly selected to evaluate the frequency of α-SMA⁺ cells out of PDGFRβ⁺ cells. Scale bars, 30 μm. (**B**) The schematic diagram showing

*Figure 4 continued on next page*

*Figure 4 continued*

the procedures of isolating REPs for flow cytometry assay. (**C**) The representative FACS plots showing the sorting strategy for PDGFRβ⁺ REPs. (**D**) *Epo* mRNA expression in PDGFRβ⁻ and PDGFRβ⁺ cells by qRT-PCR. (**E**) Western blotting showing protein levels of mitochondrial membrane proteins (TIMM23, COXIV, and TOMM20) and autophagic proteins LC3 in FACS-sorted PDGFRβ⁺ REPs from WT and *Fundc1⁻/⁻* mice under the treatment of hypoxia (1% $O_2$). (**F**) The quantification of the signal intensity of indicated mitochondrial membrane proteins (TIMM23, COXIV and TOMM20) and ratio of LC3II/I from (**E**). (**G**) Mitochondrial mass examined by using MitoTracker Green in FACS-sorted PDGFRβ⁺ REPs in WT and *Fundc1⁻/⁻* mice with or without PHZ treatment. (**H**) Flow cytometry analysis of mitophagy in REPs of mt-Keima/WT and mt-Keima/*Fundc1⁻/⁻* mice after 48 hr of PBS or PHZ treatment. (**I**) mtDNA copy number in FACS-purified REPs in WT and *Fundc1⁻/⁻* mice with or without PHZ treatment. (**J, K**) Mitochondrial membrane potential detected by using TMRM (**J**) or ROS level (**K**) measured by using MitoSOX in FACS-purified REPs of WT or *Fundc1⁻/⁻* mice after 48 hr of PBS or PHZ treatment. (**L, M**) Relative mRNA levels of proinflammatory cytokines *Il1b, Il6* (**L**) and *Epo* (**M**) in sorted PDGFRβ⁺ REPs in WT or *Fundc1⁻/⁻* mice after 48 hr of PBS or PHZ treatment, respectively. For each experiment, n = 3–5 mice for each group. Individual mice are represented by symbols. Data shown are representative of at least three independent experiments. Similar results were found in each experiment. All data are mean ± SEM; *$p<0.05$, **$p<0.01$. Statistical significance was analyzed by using the two-tailed unpaired Student's *t*-test.

The online version of this article includes the following source data and figure supplement(s) for figure 4:

**Source data 1.** Inflammation and *Epo* expression in REPs.
**Figure supplement 1.** *Fundc1* depletion impairs mitophagy and EPO production in REPs treated with PHZ *in vitro*.
**Figure supplement 1—source data 1.** Mitophagy and EPO production in REPs treated with PHZ in vitro.

To examine the effects (direct or indirect) of PHZ on EPO production in *Fundc1⁻/⁻* REPs, we FACS-sorted PDGFRβ⁺ REPs from the kidneys of mito-Keima/WT and mito-Keima/*Fundc1⁻/⁻* mice and treated them with various concentrations of PHZ *in vitro* for 12 hr (***Figure 4—figure supplement 1A***). Although a low dosage of PHZ (10 µM) had little effect on mitophagy (***Figure 4—figure supplement 1B,C***) and EPO production (***Figure 4—figure supplement 1D***), a high dosage (100 µM) of PHZ markedly impaired mitophagy and reduced EPO expression (***Figure 4—figure supplement 1D***) in *Fundc1⁻/⁻* REPs when compared with their WT counterparts. Thus, the high dosage of PHZ might have, at least in part, a direct impact on the *Fundc1⁻/⁻* REPs.

Consistent with and likely the result of the retention of the damaged mitochondria, we observed the accumulation of mtDNA copy number, reduction of mitochondrial membrane potential, and elevation of mitochondrial ROS level in PHZ-*Fundc1⁻/⁻* REPs (***Figure 4I–K***). At the molecular level, the level of proinflammatory cytokines, such as *Il1b* and *Il6*, was elevated (***Figure 4L***). Under such an inflammatory milieu, *Epo* production in PHZ-*Fundc1⁻/⁻* REPs was reduced (***Figure 4M***). These results suggest the following scenario: the reduction of EPO generation in PHZ-*Fundc1⁻/⁻* REPs is caused by deficient mitophagy, which in turn triggers the inflammatory responses and induces the fibrotic transformation of REPs. Thus, FUNDC1-mediated mitophagy plays a protective role in REPs under stresses.

### *Fundc1* deletion results in renal fibrosis

We next examined the potential link between mitochondrial quality in REPs and renal fibrosis and reduced RBCs. To achieve this goal, we applied a commonly used renal disease model, unilateral ureteral obstruction (UUO) mice that manifest inflammatory fibrosis (***Guo et al., 1999***; ***Inoue et al., 2010***; ***Figure 5A***). As shown in ***Figure 5B***, Masson's trichrome staining of the renal sections from UUO-*Fundc1⁻/⁻* mice revealed more positive blue staining of collagenous fibers, indicating that renal fibrosis was exacerbated with *Fundc1* ablation. Consistently, fibrotic markers, such as *Acta2* (encoding αSma) and collagen type 1 alpha one chain (*Col1a1*), were also significantly upregulated in kidneys of UUO-*Fundc1⁻/⁻* mice, while little difference was seen in the *Fundc1⁻/⁻* control (***Figure 5C***). In addition, the mitochondrial content was greatly reduced in kidneys of WT mice with UUO induction, as revealed by immunoblotting analysis of the mitochondrial proteins of TOMM20, COXIV, TIMM23, and FUNDC1 itself (***Figure 5D***). In contrast, mitochondrial elimination was partially blocked in the kidneys of UUO-*Fundc1⁻/⁻* mice (***Figure 5D***).

Furthermore, the level of autophagic genes, such as P62 and LC3-II, showed a trend of reduction in UUO-*Fundc1⁻/⁻* renal cells, suggesting the accumulation of dysfunctional mitochondria due to deficient autophagy (***Figure 5D***). Proinflammatory cytokines of *Il6* and *Il1b* were upregulated in the UUO-*Fundc1⁻/⁻* renal cells, likely as a consequence of defective mitochondrial clearance (***Figure 5E***).

We next focused on mitophagy in the REPs of UUO-*Fundc1⁻/⁻* mice. We found that *Fundc1* deletion induced an elevation of the mitochondrial ROS level (***Figure 5F***) and an alteration of the

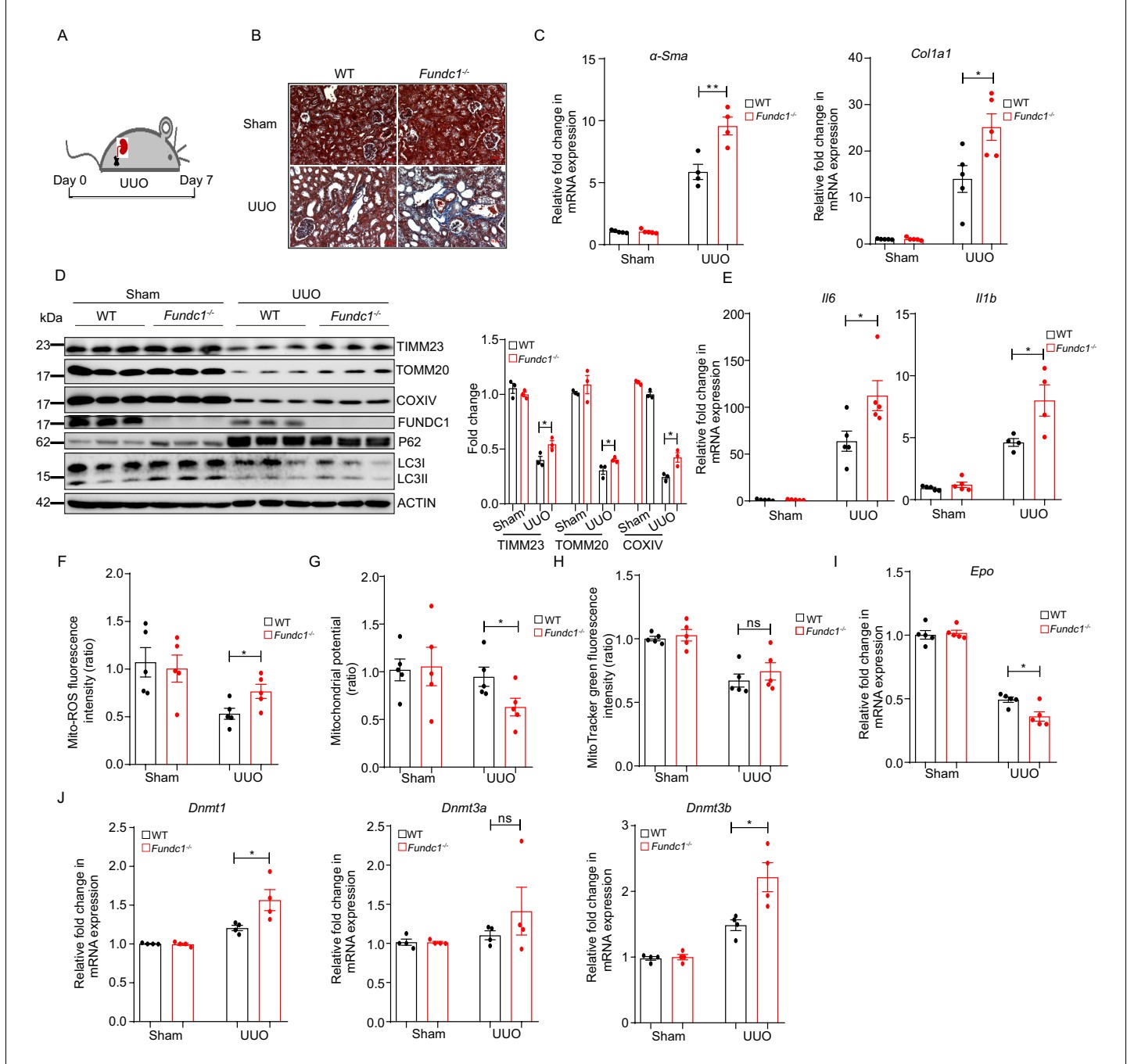

**Figure 5.** More severe renal fibrosis and increased inflammation in *Fundc1*[-/-] mice with UUO model. (**A**) Schematic diagram showing the generation of UUO pathological model from WT and *Fundc1*[-/-] mice. (**B**) Masson's trichrome staining showing collagenous fiber (blue staining) of the renal section from UUO-*Fundc1*[-/-] and WT mice. (**C**) The expression of fibrosis related genes, *α-Sma* and *Col1a1*, in kidneys of sham or UUO-induced WT and *Fundc1*[-/-] mice. (**D**) Western blotting of the protein levels of mitochondrial membrane proteins (TOMM20, TIMM23, COXIV, and FUNDC1) and autophagy associated proteins (LC3 and P62) in kidney of sham, UUO-induced WT and *Fundc1*[-/-] mice. (**E**) The mRNA expression levels of proinflammatory cytokines *Il6* and *Il1b* in kidney of sham, UUO-induced WT and *Fundc1*[-/-] mice. (**F–H**) Flow cytometry analysis of mitochondrial ROS level by MitoSOX (**F**), membrane potential by TMRM (**G**) and mitochondrial mass by MitoTracker Green (**H**) in FACS sorted, PDGFRβ+ REPs of kidneys in Sham, UUO-induced WT and *Fundc1*[-/-] mice, respectively. (**I**) Relative mRNA levels of *Epo* in sham, UUO-induced WT and *Fundc1*[-/-] mice. (**J**) The mRNA expression of DNA methyltransferase *Dnmt1*, *Dnmt3a*, and *Dnmt3b* in sham, UUO-induced WT and *Fundc1*[-/-] mice. For each experiment, n = 3–5 mice for each group. Individual mice are represented by symbols. Data shown are representative of at least three independent experiments. Similar results were found in each experiment. All data are mean ± SEM; *p<0.05; ns: no statistical significance. Statistical significance was analyzed by using the two-tailed unpaired Student's *t*-test.

*Figure 5 continued on next page*

*Figure 5 continued*

The online version of this article includes the following source data for figure 5:

**Source data 1.** Fibrosis and inflammation in UUO model.

mitochondrial membrane potential (*Figure 5G*) in REPs of UUO-*Fundc1*$^{-/-}$ mice, with minor changes in mitochondrial mass (*Figure 5H*). As a result of the accumulation of dysfunctional mitochondria in REPs, *Epo* transcription was significantly impaired in *Fundc1*$^{-/-}$ mice (*Figure 5I*). Thus, the reduction of EPO production in *Fundc1*$^{-/-}$ REPs may result from active inflammation caused by abnormal mitophagy.

We next asked how aberrant mitochondrial dysfunction resulted in the suppression of *Epo* transcription. Given that the dysregulation of mitochondrial metabolism may alter the transcription program epigenetically (*Matilainen et al., 2017*), we then examined the expression of epigenetic regulators, such as DNA methyltransferases *Dnmt1*, *Dnmt3a*, and *Dnmt3b*, in REPs. We found that these DNA methyltransferases were upregulated in UUO-*Fundc1*$^{-/-}$ REPs when compared with their WT counterparts (*Figure 5J*). Thus, impaired *Epo* transcription might be associated with the changes in epigenetic modifications.

### *Fundc1* deletion aggravates renal anemia

To confirm the notion that *Fundc1* deficiency contributes to renal anemia, we utilized cisplatin in *Fundc1*$^{-/-}$ mice to induce renal anemia (*Figure 6A*). Cisplatin is a widely used chemotherapeutic drug with notorious toxicity in the kidneys, leading to renal anemia (*Kuzur and Greco, 1980*; *Wood and Hrushesky, 1995*). We found that cisplatin-treated *Fundc1*$^{-/-}$ mice showed a fewer number of RBCs, lower HGB levels in the peripheral blood (*Figure 6B*), and a lower serum EPO level (*Figure 6C*) than in cisplatin-treated WT mice. Western blotting assay of mitochondrial membrane proteins (e. g. TOMM20 and COXIV) and autophagic proteins (e.g. P62, LC3II/I) revealed that *Fundc1* ablation impaired the clearance of damaged mitochondria (*Figure 6D*) and caused elevated inflammatory signals (*Figure 6E*) in the kidneys of cisplatin-treated *Fundc1*$^{-/-}$ mice. More importantly, deficient mitophagy was pronounced in REPs, as revealed by the direct assessment of the encapsulated mitochondria in PDGFRβ$^{+}$ cells in cisplatin-treated mito-Keima/*Fundc1*$^{-/-}$ mice (*Figure 6F,G*). We also detected increased mitochondrial mass, dysregulated mitochondrial membrane potential (indicated by TMRM), and enhanced mitochondrial ROS levels in the PDGFRβ$^{+}$ REPs in cisplatin-*Fundc1*$^{-/-}$ mice (*Figure 6H–J*). As a consequence of the abnormal mitophagy, enhanced inflammation (e.g. *Il6* and *Tnfa*) and reduced EPO generation were found in PDGFRβ$^{+}$ REPs in cisplatin-*Fundc1*$^{-/-}$ mice (*Figure 6K,L*). Thus, *Fundc1* deficiency impairs the clearance of damaged mitochondria and activates inflammation, which results in decreased EPO production in PDGFRβ$^{+}$ REPs and, ultimately, aggravates the renal anemia (*Figure 6M*).

## Discussion

### FUNDC1 in stress versus steady-state erythropoiesis

Our current study showed that *FUNDC1* is required for EPO production upon PHZ-induced stresses (*Figure 6M*). Specifically, we found that FUNDC1-mediated mitophagy is essential for EPO production in REPs during stress-induced kidney injuries and that damaged mitochondria accumulate in *Fundc1*$^{-/-}$ REPs as a result of impaired mitophagy. Consequently, an elevation of the ROS level and the release of mtDNA from damaged mitochondria in REPs incur inflammatory responses by enhancing the expression of proinflammatory cytokines including *TNFa*, *IL6*, and *IL1b*. These changes in the inflammatory milieu promote myofibroblastic transformation of REPs and the concomitant loss of EPO production capacity and, ultimately, lead to renal anemia. To the best of our knowledge, we provide the first link between mitophagy and the inflammatory state in REPs, demonstrating that mitophagy and mitochondrial quality control play protective roles in REP function and renal anemia.

Additionally, it is somewhat surprising that FUNDC1 is dispensable for the programmed mitochondrial elimination in the final step of erythroid maturation, whereas another mitophagy receptor BNIP3L is required for the removal of mitochondria in reticulocytes during terminal erythroid differentiation (*Schweers et al., 2007*; *Sandoval et al., 2008*). However, during cardiac lineage

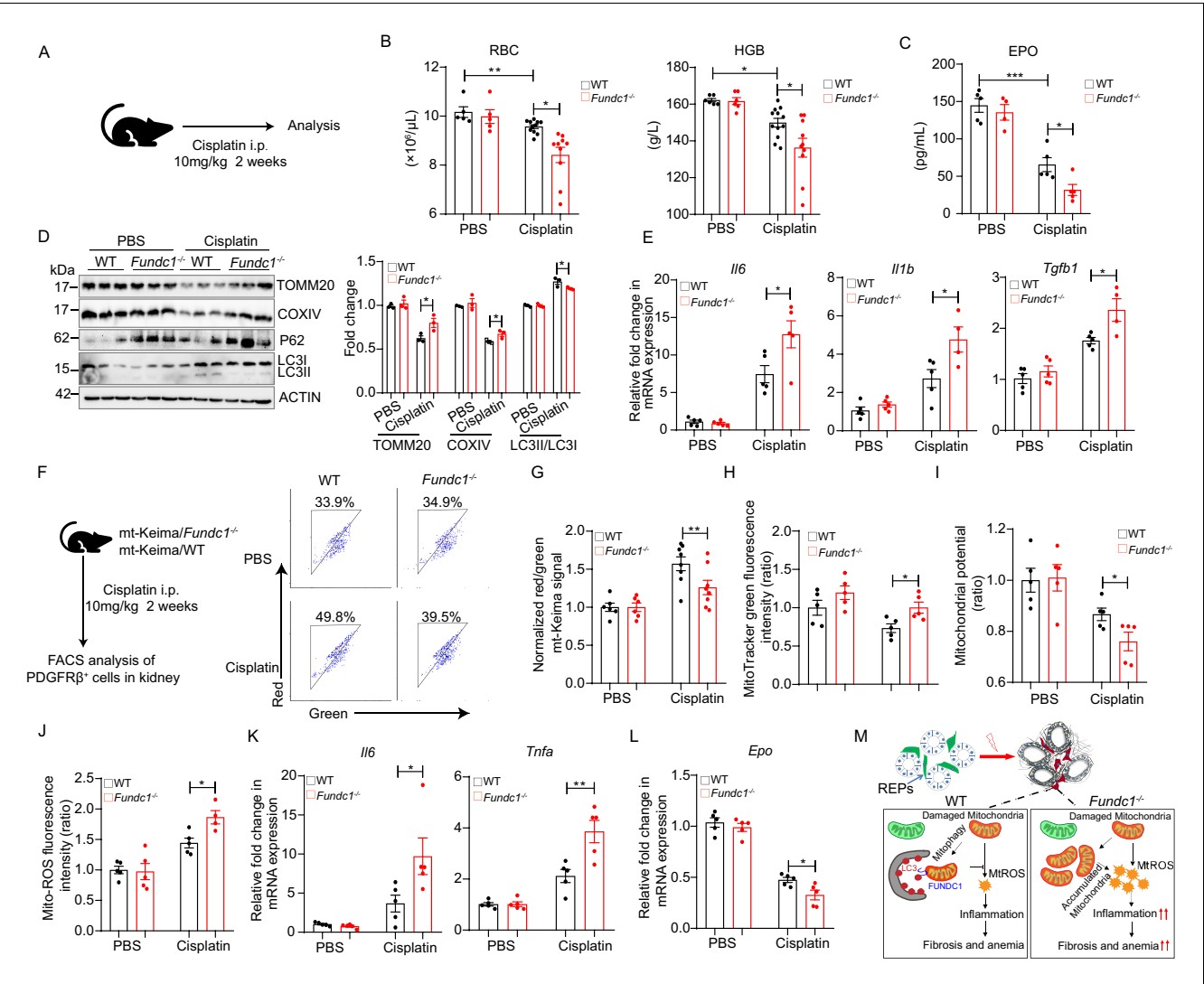

**Figure 6.** Impaired mitophagy and reduced EPO production in *Fundc1⁻/⁻* mice during cisplatin-induced renal anemia. (A) Schematic diagram showing the generation of renal anemia disease model by cisplatin. (B) Hemogram parameters of red blood cell (RBC) counts and hemoglobin (HGB) in the peripheral blood of WT controls and *Fundc1⁻/⁻* mice after PBS or cisplatin treatment. (C) The serum EPO concentration in *Fundc1⁻/⁻* and WT mice after PBS or cisplatin treatment. (D) Western blotting (left) and the corresponding quantification (right) showing the protein levels of mitochondrial membrane proteins (TOMM20 and COXIV) and autophagic proteins (P62 and ratio of LC3II/I) in kidneys of WT and *Fundc1⁻/⁻* mice with PBS or cisplatin treatment. (E) Relative mRNA levels of *Il6*, *Il1b*, and *Tgfb*1 in kidneys of WT and *Fundc1⁻/⁻* mice with or without cisplatin treatment. (F, G) Flow cytometry analysis of mitophagy in PDGFRβ⁺ REPs of mt-Keima/WT and mt-Keima/*Fundc1⁻/⁻* mice after PBS or cisplatin treatment. The representative FACS plots (F) and the quantification (G) are shown. (H) Mitochondrial mass detected by using MitoTracker Green via FACS in PDGFRβ⁺ REPs of WT or *Fundc1⁻/⁻* mice after PBS or cisplatin treatment. (I) Mitochondrial membrane potential detected by using TMRM by FACS in PDGFRβ⁺ REPs of WT or *Fundc1⁻/⁻* mice after PBS or cisplatin treatment. (J) Mitochondrial ROS level detected by using MitoSOX via FACS in PDGFRβ⁺ REPs of WT or *Fundc1⁻/⁻* mice after PBS or cisplatin treatment. (K) Relative mRNA levels of proinflammatory cytokines *Il6* (left) and *Tnfa* (right) in sorted PDGFRβ⁺ REPs in WT or *Fundc1⁻/⁻* mice after PBS or cisplatin treatment. (L) Relative mRNA levels of *Epo* in sorted PDGFRβ⁺ REPs in WT or *Fundc1⁻/⁻* mice after PBS or cisplatin treatment, respectively. (M) A hypothetic model depicting the role of FUNDC1 in REPs. FUNDC1-mediated mitophagy is required for mitochondrial steady-state homeostasis. However, under stresses, the damaged mitochondria are accumulated in *Fundc1⁻/⁻* REPs due to impaired mitophagy. Consequently, elevated ROS levels from these damaged mitochondria of REPs incur inflammatory responses by enhancing the expression of proinflammatory cytokines *TNFa*, *IL6* and *IL1b*, which in turn promote myofibroblastic transformation of REPs, resulting in the loss of EPO generation and subsequently anemia. For each experiment, n = 3–5 mice for each group. Individual mice are represented by symbols. Data shown are representative of at least three independent experiments. Similar results were found in each experiment. All data are mean ± SEM; *p<0.05, **p<0.01, ***p<0.001. Statistical significance was analyzed by using the two-tailed unpaired Student's *t*-test.

The online version of this article includes the following source data for figure 6:

**Source data 1.** Mitophagy and EPO production during cisplatin-induced renal anemia.

differentiation, both FUNDC1 and BNIP3L are required for the remodeling of the mitochondrial network (*Lampert et al., 2019*). Thus, it appears that the function of FUNDC1 and BNIP3L is highly context- and cell-type dependent.

## Mitochondrial quality and EPO production in REPs

Renal EPO production is central for erythropoiesis. It has been well-established that HIF2α plays essential roles in hypoxia-induced *EPO* gene expression (*Gruber et al., 2007*; *Semenza et al., 1991*). In this study, we found that the expression of HIF2α is largely unaffected by *Fundc1* deletion, indicating that reduced EPO production in *Fundc1$^{-/-}$* mice is likely to be unrelated to the oxygen-sensing mechanism. Instead, the low yield of EPO may be due to the reduced number of REPs and/or compromised function during REP transformation to myofibroblasts. Nonetheless, we found that the fraction of PDGFRβ$^+$ REPs is unchanged in *Fundc1$^{-/-}$* mice when compared with the WT controls under stresses (data not shown). However, in the purified REPs from *Fundc1$^{-/-}$* mice, we showed that the increase in the mitochondrial ROS levels, as a result of defective mitophagy, provokes inflammatory responses, including the upregulation of TNFa, IL6 and IL1b in REPs, and subsequently lead to fibrotic transformation. Although prior studies demonstrated that activation of inflammatory signals, such as NFκB (*Souma et al., 2013*; *Liu et al., 2012b*), TNFa (*Frede et al., 1997*), IL1b (*Frede et al., 1997*), IL6 (*Stenvinkel et al., 2005*), repress EPO production in REPs and promote their transition to myofibroblasts (*Guo et al., 1999*; *Inoue et al., 2010*; *Souma et al., 2013*; *Ernandez and Mayadas, 2009*), in this study we provide the first link between deficient mitophagy and EPO production in REPs with the induction of inflammation as the intermediate step. We argue that inefficient mitophagy caused by *Fundc1* deletion results in the loss of EPO production in REPs, resembling the clinical manifestations of renal anemia. Therefore, by selectively removing the damaged mitochondria, mitophagy represents an essential element of mitochondrial quality control in response to renal injury or anemia caused by UUO, PHZ, or cisplatin. Although PDGFRβ is the most frequently used surface marker to identify or isolate REPs (*Gerl et al., 2016*; *Greenwald et al., 2019*; *Broeker et al., 2020*), PDGFRβ$^+$ cells are heterogenous and dynamic (*Pan et al., 2011*). Therefore, the role of mitophagy in REPs awaits future investigation.

In addition, defective mitophagy might also result in metabolic reprogramming, which is widely observed in renal epithelial cells or myofibroblast transformation (*Xu et al., 2019*; *Zhao et al., 2017*; *Livingston et al., 2019*), and the alteration in mitochondrial metabolism may regulate EPO production epigenetically. In line with this speculation, DNMTs, including DNMT1, DNMT3a, and DNMT3b, are upregulated, suggesting that altered epigenetic modifications might be responsible for the reduced EPO production in *Fundc1$^{-/-}$* REPs, consistent with previous observations in mouse myofibroblast-transformed REPs (*Souma et al., 2013*; *Chang et al., 2016*) and the cell lines generated from EPO-producing cells without the capabilities of EPO production (*Yin and Blanchard, 2000*; *Steinmann et al., 2011*; *Sato et al., 2019*). Thus, our current study further highlights the promise of applying demethylating agents to the restoration of EPO expression in patients with renal anemia.

## Mitochondrial dysfunction, renal fibrosis, and renal diseases

Renal anemia is one of the most common complications among various chronic kidney diseases (*Quaggin and Kapus, 2011*). Renal anemia adversely affects the quality of patients' life and is strongly associated with poor clinical outcomes (*Collins et al., 1998*). The treatment of renal anemia has long been dependent on erythropoiesis-stimulating agents and iron supplement in chronic kidney diseases, raising the concerns of increased patients' death and cardiovascular risk (*Solomon et al., 2010*; *Bohlius et al., 2009*). In the past decades, tremendous efforts have been focused on understanding the cellular basis and molecular regulatory mechanism of REPs, in order to offer more efficient and safer treatment options.

Myofibroblastic transformation of REPs, which causes REPs to lose their EPO-producing ability (*Koury, 2014*), is one of the major sources of myofibroblasts contributing to renal fibrosis (*Souma et al., 2013*). Despite the established correlation between inflammation and REP fibrotic conversion, it is largely unclear how the upstream pathological inflammatory pathways are triggered in REPs and how mitochondrial quality control is linked to REPs in chronic kidney disorders or injuries. In this study, we showed that the kidney of *Fundc1$^{-/-}$* mice displays a more severe fibrotic phenotype in UUO model. Concomitant with the inability of REPs to secret EPO, these cells undergo

myofibroblastic conversion. Thus, receptor-mediated mitophagy protects against renal anemia and possible renal fibrosis. In addition, we found that *Fundc1* is abundantly and widely expressed in the kidney. It is therefore highly likely that FUNDC1-mediated mitophagy might also play crucial roles in other types of renal cells besides REPs, and that these cells might also contribute to the death of *Fundc1*[-/-] mice under PHZ stresses. For example, during this study, it was reported that FUNDC1-dependent mitophagy in tubule cells plays protective roles in ischemic acute kidney injury (*Wang et al., 2020*). It appears that targeting FUNDC1-mediated mitophagy should have pleiotropic therapeutic impacts by 'killing two birds with one stone'. In renal fibrosis, enhanced mitophagy might not only ameliorate anemia by attenuating inflammation in REPs, but also alleviate fibrosis by preserving and protecting the mitochondrial quality from other fibrogenic contributing cells. These exciting possibilities should be examined in future investigations.

## Materials and methods

### Mouse experiments

Mice were maintained in the animal core facility of College of Life Sciences, Nankai University, Tianjin, China. All experiments involving animals were reviewed and approved by the Animal Care and Use Committee of Nankai University and were performed in accordance with the university guidelines. All mice in this study were of the C57BL/6 background. *Fundc1* whole-body knockout mice (*Fundc1*[-/-]) were generated as previously described (*Zhang et al., 2016*). The WT and *Fundc1*[-/-] mice around 8–10 weeks old were injected intraperitoneally with phenylhydrazine (PHZ, cat# P26252, Sigma, 100 mg/kg body weight) to induce acute hemolysis anemia. For the complementary experiments, after 12 hr of PHZ treatment, EPO (cat# 46130, Epiao, 1000 IU/kg body weight) was injected subcutaneously into PHZ-*Fundc1*[-/-] and PHZ-WT mice three times with an interval of 12 hr (*Nai et al., 2016*; *Patel et al., 2004*; *Kwak et al., 2020*). These mice were sacrificed at 48 hr after PHZ treatment for further analyses. We induced pure anemia via phlebotomy. In detail, 500 μL of blood was taken from the tail vein of mice for three consecutive days, and they were sacrificed for further analyses (*Casanovas et al., 2013*; *Kautz et al., 2014*). To generate the unilateral ureteral obstruction (UUO) mice model, the left ureter was obstructed by ligation with a suture or by the application of a ligating clip in adult mice. Kidneys were harvested on day 7 after unilateral ureteral obstruction for further analysis, as previously described (*Hesketh et al., 2014*). We also injected cisplatin (cat# S1166, Selleckchem, 10 mg/kg body weight) intraperitoneally into WT and *Fundc1*[-/-] mice around 8–10 weeks old to induce renal anemia. These mice were sacrificed at day 14 after cisplatin injection for further analyses (*Kuzur and Greco, 1980*; *Wood and Hrushesky, 1995*; *Wang et al., 2018*). Mitochondrial-targeted transgenic mice (mt-Keima) were crossed with *Fundc1*[-/-] mice to breed the *mt-Keima/Fundc1*[-/-] mice.

### Flow cytometry and cell sorting

Primary antibodies used for flow cytometry were as follows: CD71 (clone RI7217, BioLegend), Ter-119 (clone TER-119, BioLegend), CD3 (clone 17A2, BioLegend), B220 (clone RA3-6B2, BioLegend), LY6G (clone 1A8, BioLegend), LY6C (clone HK1.4, BioLegend), F4/80 (clone BM8, BioLegend), Sca1 (clone D7, BioLegend), c-Kit (clone 2B8, BioLegend), CD34 (clone RAM34, eBioscience), CD16/32 (clone 93, BioLegend), PE/Cy7-Streptavidin (cat# 405206, BioLegend), Biotin-CD140b (clone APB5, BioLegend), and Brilliant Violet 605 Streptavidin (cat# 405229, BioLegend). The erythroid cells were stained with CD71 and Ter119; hematopoietic stem and progenitor cells were immunophenotypically labeled as LSK (Lin⁻Sca1⁺c-Kit⁺) and LK (Lin⁻c-Kit⁺); common myeloid progenitor (CMP) cells were labeled as Lin⁻Sca1⁺c-Kit⁺CD34⁺CD16/32⁻; granulocyte-macrophage progenitor (GMP) cells were labeled as Lin⁻c-Kit⁺CD34⁺CD16/32⁺; and megakaryocyte-erythroid progenitor (MEP) cells were labeled as Lin⁻c-Kit⁺CD34⁻CD16/32⁻. Neutrophils, T cells, B cells, and monocytes were marked as Gr1⁺, CD3⁺, B220⁺, and F4/80⁺, respectively.

Fluorescence-activated Cell Sorting (FACS) was conducted as previously described (*Liu et al., 2018*). Briefly, the cells were collected, washed, and resuspended in ice-cold phosphate-buffered saline (PBS) supplemented with 2% FBS. For each assay, approximately $10^6$ cells were stained with antibodies at 4°C for 30 min in the dark. Cells were then washed and resuspended in 300 μL ice-cold PBS–FBS and subjected to flow cytometry analysis on a FACS LSRII flow cytometer (BD Biosciences).

## Mitochondrial assays

The mitochondrial membrane potential was analyzed using tetramethylrhodamine methyl ester perchlorate (TMRM, cat# T668; Invitrogen) staining, following the manufacturer's instructions. Briefly, cells were stained with TMRM at a final concentration of 200 nM for 30 min at 37°C and then stained with Ter119 and CD71 antibodies before flow cytometry. MitoTracker Green (MTG; cat# M7514, Thermo Fisher Scientific) was used to analyze mitochondrial content following the manufacturer's instructions. Cells were stained with MTG at a final concentration of 200 nM for 30 min at 37°C and then stained with Ter119 and CD71 antibodies before flow cytometry analysis. The mitochondrial ROS level was analyzed with the MitoSOX red mitochondrial superoxide indicator (cat# M36008, Thermo Fisher Scientific) according to the manufacturer's instructions. Cells were stained with MitoSOX red mitochondrial superoxide indicator at a final concentration of 5 µM for 30 min at 37 °C and then stained with Ter119 and CD71 antibodies before flow cytometry analysis. Data were analyzed with FlowJo (version 7.6.1).

## *Ex vivo* maturation of reticulocytes

FACS-sorted Ter119$^+$CD71$^+$ reticulocytes from the peripheral blood were cultured for 3 days in IMDM medium (I3390, Sigma) containing 2 mM L-glutamine (cat# 25030, Gibco) and 100 U penicillin-streptomycin (cat# DE17-602E, Lonza), 30% FBS (cat# 10270, Gibco), 1% BSA (cat# A1595, Sigma), and 0.001% monothioglycerol. In the final stage, 10 µM FCCP (HY-100410, MCE) was added to the cultured reticulocytes.

## Western blotting

Western blotting was performed as previously described (*Liu et al., 2018*). Briefly, cells were lysed in Laemmli sample buffer (cat# 161–0737, Bio-Rad) before SDS–PAGE. Primary antibodies used for western blotting were as follows: anti-FUNDC1 (home-made) (*Liu et al., 2012a*), anti-LC3 (cat# L8918, Sigma), anti-TIMM23 (cat# 611222, Biosciences), anti-TOMM20 (cat# 612278, Biosciences), anti-STAT3 (cat# 9139, Cell Signaling Technology), anti-phosphorylated STAT3 (cat# 9134, Cell Signaling Technology), anti-STAT5 (cat# 94205, Cell Signaling Technology), anti-phosphorylated STAT5 (cat# 4322, Cell Signaling Technology), anti-HIF2α (cat# NB100-122, Novus Biologicals), anti-βACTIN (cat# 8H10D10, Cell Signaling Technology), and anti-αTUBULIN (cat# ab52866, Abcam). Horseradish peroxidase-conjugated secondary antibodies were used, and signals were detected using an ECL kit (cat# WP20005, Invitrogen).

## ELISA of cytokines

The sera collected from the mice with distinct genotypes were stored at −80°C. Inflammatory cytokines were measured using Mouse Inflammation Panel (13-plex) with V-bottom Plate (cat# 740446, BioLegend), according to the manufacturer's instructions.

The concentrations of EPO were measured using ELISA kit (cat# ab119593, Abcam). Briefly, the standards and samples were added to the wells and incubated for 90 min at 37°C. After samples were washed, a biotinylated goat polyclonal antibody specific for EPO was supplemented. Finally, avidin-biotin-peroxidase complex was incubated for 30 min for the enzymatic reaction. The signal was assessed by using a microplate reader (Synergy H4, BIO-TEK).

## Histological analysis of the renal tissue sections

Kidneys were fixed in 4% paraformaldehyde overnight at room temperature and dehydrated in 20% sucrose overnight on the next day. The paraffin-embedded tissues were then cut into 4 µm thick sections. For staining, the slides were dewaxed and incubated with mouse anti-α-Smooth Muscle-Cy3 antibody (cat# 1A4, Sigma, 1:200) and anti-PDGFRβ antibody (cat# 28E1, Cell Signaling Technology, 1:200) at 4°C overnight, followed by staining with an Alexa Fluor-conjugated secondary antibody. Fluorescent images were obtained using a confocal microscope (UltraVIEW Vox, PerkinElmer).

A TUNEL assay was used to detect apoptotic cells (cat# 12156792910, Roche Applied Science), according to the manufacturer's instructions. In brief, tissue sections were deparaffinized and permeabilized with 0.1 M sodium citrate and incubated with a TUNEL reaction mixture for 1.5 hr at 37°C in a humidified, dark chamber. For quantification, approximately 20 fields (200×) were randomly selected to evaluate the TUNEL-positive cells.

## Measurement of cytosol-releasing mitochondrial DNA (mtDNA) in kidney

Fifty to 100 mg renal tissues were collected and washed with PBS. The tissues were then cut into small pieces before the addition of 10 volumes of pre-chilled mitochondrial separation reagent A and homogenized for 10–15 times. The produced homogenates were centrifuged at 600 g at 4°C for 5 min to remove the cell nuclei, while the supernatant was centrifuged at 12,000 g at 4°C for 10 min to further remove the intact mitochondria. The remaining cytosolic free mtDNA was purified and quantified by using qRT-PCR with mtDNA-specific primers. The primers are listed in *Supplementary file 2*.

## RNA extraction and qRT-PCR

Total RNA was extracted with TRIzol (cat# 15596026, Invitrogen) and reversely transcribed using TransScript II One-Step gDNA Removal and cDNA Synthesis SuperMix (cat# AH311-03, TransGen Biotech). The qRT-PCR was completed using the SYBR Green Master Mix kit (cat# A25742, Thermo Fisher Scientific), and the assay was conducted with the QuantStudio5 Real-Time PCR System (Life Technologies). The primers are listed in *Supplementary file 2*.

## Isolation of PDGFRβ⁺ REPs

Kidneys were decapsulated, diced on ice, and then incubated in RPMI 1640 medium containing 0.5 mg/mL liberase (cat# 423610, Roche) and 100 U/mL DNase I (cat# D8071, Solarbio) at 37°C for 45 min, followed by supplementation of 0.1% bovine serum albumin to stop the enzymatic reaction. After filtration and red blood cell lysis, the remaining single cells were incubated with PDGFRβ⁺ antibody for 20 min at 4°C before flow cytometry analysis.

## RNA-sequencing and data analysis

Total RNA from FACS-purified R2 cells of the spleen of PHZ-treated WT (n = 3) and *Fundc1$^{-/-}$* mice (n = 3) were extracted with TRIzol (cat# 15596026, Invitrogen). Library construction and data processing were performed by Novogene (Beijing, China), as described previously (*Liu et al., 2018*). In brief, quality control was conducted by using FastQC (http://www.bioinformatics.babraham.ac.uk/projects/fastqc/), and clean reads were aligned by HISAT2 (*Pertea et al., 2016*) to the mouse reference genome GRCm38 (https://www.gencodegenes.org/) of gencode.vM17 (https://www.gencodegenes.org/). Next, the expression of genes was quantified by using StringTie (*Pertea et al., 2016*) with default parameters, and differentially expressed genes (DEGs) were identified by using DESeq2 (*Love et al., 2014*) within R package. Genes with fold change > 1.5 and p- value < 0.05 were considered as significant DEGs.

## Functional annotation of DEGs

Functional annotation of significant DEGs was performed using 'Investigate Gene Sets' in GSEA (https://www.gsea-msigdb.org/gsea/msigdb/annotate.jsp). We selected KEGG (Kyoto Encyclopedia of Genes and Genomes) as our target database, while the p-values were adjusted using the Benjamini and Hochberg method with significant enrichment as FDR < 0.05.

## Gene set enrichment analysis

The JAK-STAT gene set derived from MSigDBwas used for GSEA (http://software.broadinstitute.org/gsea/index.jsp). All of the expressed genes (average FPKM > 0.01) from different groups were used as the input dataset. The numbers of permutations were set to 1000. p-values were corrected for multiple testing methods, and the threshold for significantly enrichment was FDR < 0.05.

## Statistical analysis

Statistical significance was calculated using the two-tailed unpaired Student's t-test unless stated otherwise. Data in bar graphs were represented as mean ± SEM, and statistical significance was expressed as follows: *p<0.05; **p<0.01; ***p<0.001; ns: no significant.

## Additional information

### Funding

| Funder | Grant reference number | Author |
| --- | --- | --- |
| National Key Research and Development Program of China | 2019YFA0508601 | Quan Chen |
| National Natural Science Foundation of China | 91849201 | Quan Chen |
| National Key Research and Development Program of China | 2019YFA0508603 | Yushan Zhu |
| National Natural Science Foundation of China | 32030026 | Yushan Zhu |
| National Natural Science Foundation of China | 81870089 | Lihong Shi |
| Chinese Academy of Medical Sciences | 2016-I2M-3-002 | Lihong Shi |
| National Key Research and Development Program of China | 2017YFA0103102 | Quan Chen |
| National Key Research and Development Program of China | 2016YFA0102300 | Quan Chen |
| National Natural Science Foundation of China | 31790404 | Quan Chen |
| National Natural Science Foundation of China | 81890990 | Lihong Shi |
| National Natural Science Foundation of China | 81700105 | Lihong Shi |
| Chinese Academy of Medical Sciences | 2019-I2M-1-006 | Lihong Shi |
| Chinese Academy of Medical Sciences | 2016-I2M-1-018 | Lihong Shi |
| Chinese Academy of Medical Sciences | 2017-I2M-1-015 | Lihong Shi |
| National Natural Science Foundation of China | 91754114 | Yushan Zhu |

The funders had no role in study design, data collection and interpretation, or the decision to submit the work for publication.

### Author contributions

Guangfeng Geng, Conceptualization, Data curation, Writing - original draft; Jinhua Liu, Data curation, Formal analysis, Validation; Changlu Xu, Data curation, Software, Formal analysis, Validation; Yandong Pei, Resources, Methodology; Linbo Chen, Resources, Validation; Chenglong Mu, Resources, Validation, Methodology; Ding Wang, Software, Formal analysis; Jie Gao, Resources, Supervision, Methodology; Yue Li, Formal analysis; Jing Liang, Resources, Supervision, Visualization; Tian Zhao, Software, Validation, Methodology; Chuanmei Zhang, Validation, Methodology; Jiaxi Zhou, Supervision, Methodology; Quan Chen, Resources, Funding acquisition, Project administration, Writing - review and editing; Yushan Zhu, Conceptualization, Data curation, Funding acquisition, Writing - review and editing; Lihong Shi, Formal analysis, Funding acquisition, Writing - original draft, Writing - review and editing

## Author ORCIDs
Guangfeng Geng (iD) https://orcid.org/0000-0002-0518-406X
Changlu Xu (iD) https://orcid.org/0000-0002-1581-7027
Yushan Zhu (iD) https://orcid.org/0000-0002-5648-0416
Lihong Shi (iD) https://orcid.org/0000-0001-8876-0802

## Ethics
Animal experimentation: Mice were maintained in the animal core facility of College of Life Sciences, Nankai University, Tianjin, China. All experiments involving animals were reviewed and approved by the Animal Care and Use Committee of Nankai University and were performed in accordance with the university guidelines (NO. 2021-SYDWLL-000410).

## Decision letter and Author response
Decision letter https://doi.org/10.7554/eLife.64480.sa1
Author response https://doi.org/10.7554/eLife.64480.sa2

## Additional files

### Supplementary files
• Supplementary file 1. Hemogram of peripheral blood.
• Supplementary file 2. Primers used for qRT-PCR.
• Transparent reporting form

### Data availability
RNA-Sequencing data is deposited at GEO Accession number GSE 158361. Information on replicates is presented in Materials and Methods as well as in figure legend. Replicate numbers are mentioned in figure legends. All data generated or analysed during this study are included in the manuscript and supporting files.

The following dataset was generated:

| Author(s) | Year | Dataset title | Dataset URL | Database and Identifier |
|---|---|---|---|---|
| Geng G | 2021 | Receptor-mediated mitophagy regulates EPO production and protects against renal anemia | https://www.ncbi.nlm.nih.gov/geo/query/acc.cgi?acc=GSE158361 | NCBI Gene Expression Omnibus, GSE158361 |

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
