## [Decision Letter]

**Acceptance summary:**

Mitochondrial dysfunction is often observed in renal and red blood cell disorders, however the role of mitochondrial quality control in renal anemia remains unclear. This study describes that FUNDC1, a mitophagy receptor, is required for EPO production upon stresses due to impaired mitochondrial quality control in renal EPO-producing cells. These findings support a new mechanism that FUNDC1-mediated mitophagy regulates EPO production under stresses to protect anemia of kidney disease.

**Decision letter after peer review:**

Thank you for submitting your article "Receptor-mediated mitophagy regulates stress erythropoiesis and confers protection against renal anemia" for consideration by *eLife*. Your article has been reviewed by 3 peer reviewers, and the evaluation has been overseen by a Reviewing Editor and Matt Kaeberlein as the Senior Editor. The reviewers have opted to remain anonymous.

The reviewers have discussed the reviews with one another and the Reviewing Editor has drafted this decision to help you prepare a revised submission.

Summary:

All the reviewers find the work of interest by analyzing the erythroid defects caused by deletion of mitophagy receptor FUNDC1 in mice. The study reveals that FUNDC1 loss results in the accumulation of impaired mitochondria in renal EPO producing cells, inflammation, myofibroblastic transformation, and reduced EPO production and anemia in experimental models of acute anemia or kidney injury. The results suggest that FUNDC1 is required for mitochondrial removal during erythropoiesis, and provide new insights into the anemia of kidney disease. The reviewers have raised several major concerns related to the strengths of the main conclusions. The complete comments are included.

Reviewer 1 also suggested that the title of the manuscript and other relevant texts need to be revised to better reflect the main findings or additional evidence is needed to support the conclusions related to stress erythropoiesis.

Essential revisions:

1. It is important to provide more convincing evidence to support the conclusions related to severe anemia or stress erythropoiesis observed in FUNDC1-deleted mice, or the major conclusions should be significantly revised.

2. Evidence is needed to show whether defective mitophagy results in reduced Epo production, thus supporting the conclusion that the defective mitophagy observed in FUNDC1-deleted kidney is responsible for the decreased Epo production and associated phenotypes. A more direct experiment is required to assess the effect of PHZ on Epo production in Fundc1-/- kidney cells. To further support the hypothesis, supplemental Epo administration should be performed to examine the effects on the abnormalities observed in the Fundc1-/- mice.

3. It is important to provide evidence for mitochondrial retention in a model for kidney disease that is accompanied by anemia. Such studies will strengthen the conclusion that FUNDC1 deficiency contributes to anemia associated with kidney disease by regulating mitophagy.

4. Given the modest effects on blood cell production, the conclusions that FUNDC1-mediated mitophagy is not required for reticulocyte maturation need to be qualified or significantly revised.

5. PHZ leads to significant reduction of Epo transcript in FUNDC1-deleted kidney cells but only a mild reduction of serum Epo level. This discrepancy should be explained.

6. Given the modest effects observed in FUNDC1-deleted mice and the limitations of the PHZ-induced acute anemia, it is important to include studies using a model for chronic anemia, and examine the effects of modest Epo deficiency effects over time. For example, phlebotomy may help to distinguish the effects of oxidative stress versus pure anemia. Repeated phlebotomy may help to assess the role of Fundc1 in chronic anemia could have a more profound effect than what was observed after a single PHZ treatment.

7. Given the limitations of analyzing p62 and LC3 expression to represent altered autophagy, additional experiments are required to support conclusions related to autophagy.

*Reviewer #1:*

The authors have contributed substantially to the understanding of FUNDC1 as a mitophagic receptor in multiple cell types and disease conditions including hypoxia-mediated mitophagy in mammalian cells, inhibition of inflammasome activation, ischemia-reperfusion injury to the heart. Here the authors turned their attention to the kidney and showed that lack of FUNDC1 results in the accumulation of impaired mitochondria in EPO producing cells (REPs), leading to an inflammatory response, a myofibroblastic transformation of REPs, resulting in reduced EPO production and anemia in two experimental models including PHZ and UUO. Work presented here indicates that FUNDC1 does not play a role in mitochondrial removal during erythropoiesis. Overall, the paper is well written and presents strong evidence showing the importance of FUNDC1-mediated mitophage in renal fibrosis-mediated anemia by reducing the Epo production. However, the paper can be improvde by addressing following major concern:

The title and manuscript's emphasis of FUNDC1-mediaetd mitopahge regulting "stress eythropoiesis" are misleading. One of major complications of renal fibrosis is anemia due to the reduced production of Epo. Thus, simply because a particular protein deficiency results in renal fibrosis and reduced EPO production does not mean that such a protein should be considered as a regulator of "stress erythropoiesis" as implied by the title and multiple places in the manuscript. The findings showed here support that FUNDC1 protects against the accumulation of impaired mitochondria, inflammation, renal fibrosis and reduced EPO production and thus the development of major complication of anemia but not stress erythropoesis. Please consider to revise the title and text to avoid misleading and confusion.

*Reviewer #2:*

The manuscript by Shi et al., analyzes the effect of deletion of mitophagy receptor FUNDC1 in mice on erythropoiesis and kidney production of erythropoietin (Epo). The authors have performed a considerable amount of work and generated a large set of data. Since the authors did not find any effect of FUNDC1 deletion on hematopoiesis or erythropoiesis, they subjected mice to erythroid stress by inducing hemolytic anemia using the established phenylhydrazine (PHZ) treatment. Mice survival in response to phenylhydrazine was reduced and attributed to severe anemia. They noticed a mild reduction in Epo production in response to phenylhydrazine. They then focused on kidney production of Epo. They confirm that FUNDC1-deleted kidney cells exhibit defective mitophagy, and PHZ treatment enhanced defective mitophagy in FUNDC1-deleted kidney cells. They attributed this defect to reduced Epo production. There are two major problems in this manuscript. One is that in contrary to the authors assertion, it is unclear whether FUNDC1-deleted mice exhibit anemia (hemoglobin is not reported in PHZ-treated mice and other RBC indices are very mild) but possibly only a reduced compensated red blood cell production in response to phenylhydrazine treatment. The other is that although the authors confirm that FUNDC1-deleted kidney cells exhibit defective mitophagy that is more pronounced in PHZ-treated animals, they do not show that reduced Epo production results from defective mitophagy. My specific comments are below:

– There is no evidence of mitochondrial retention in the spleen, BM and PB. The authors use CD71 as a marker but it is unclear on which cells.

– Although the survival in response to phenylhydrazine is reduced which was achieved by using 100mg/kg (instead of the usual dose of 50mg/kg) and including 30 mice (p=0.04), there was no severe anemia. In fact Hb levels are normal and the slight reduction in red blood cell indices suggest that there was a mild reduced red blood cell production that did not reach the definition of anemia. So, in the absence of anemia the claim that Fundc1-mediated mitophagy is not required for reticulocyte maturation is not accurate.

– It remains unclear what animals die from.

– The authors FACS-sort reticulocytes 5 days post phenylhydrazine. This is an interesting experiment. What conditions for the FACS sorting do they use?

– From Figure S2 it seems that the authors FACS-sorted immature erythroblasts but not reticulocytes, the data on mitochondria in these cells also seems to be from immature erythroid cells. The authors should clarify how they distinguish reticulocytes from immature erythroblasts.

– The authors state: "Together, our results suggest that FUNDC1- mediated mitophagy is

dispensable for programmed mitochondrial elimination in reticulocytes during

erythroid maturation." It is difficult to assess the validity of this statement given that it is unclear which cells were used in these experiment.

– The authors examine differential gene expression in R2 erythroblast subpopulation between cells from mice subjected to PHZ treatment or not. They use RNA-Seq analysis of R2 and identify among others JAK-STAT pathway to be differentially expressed in PHZ treated Fundc1-/- spleens. They conclude that JAK-STAT is hypoactivated in Fundc1-/- R2 cells. Given that the reduced RBC production seem to be compensated (see R3, R4) does not seem to be highly affected this result seems to indicate compensated JAK-STAT hypoactivation.

– How do the authors explain that Epo reduction in Fundc1-/- cells has an effect solely on R2 spleen cells?

– The significant reduction of Epo transcript in kidney cells in response to PHZ is interesting but how does this correlate with a mild reduction of Epo in PB?

– To determine the role of defective mitophagy on Epo production, the authors selected PDGFRB+ and – cells and subjected to hypoxia. A more direct experiment is to assess the effect of PHZ on Epo production in Fundc1-/- kidney cells instead they have measured mitochondrial mass which does not address their hypothesis.

– How are experiments in 4M and 3A different? What is missing is the effect of PHZ on Fundc1-/- PDGFRB+ cells?

*Reviewer #3:*

The authors show that loss of the mitophagy receptor Fundc1 in mice causes defects in stress erythropoiesis by impairing the production of Epo by the kidney, with defective mitophagy of Epo-producing cells. This is interesting work that has the potential to provide insights into the anemia of kidney disease. However, the overall effects seem to be fairly modest and several of the studies must be investigated further to support the claims to a reasonable level. Notably, modest impairment of Epo production could lead to significant anemia over time, but this is not shown. Perhaps the most important problem is that study does not examine a model for anemia of kidney disease.

1. Following PHZ administration to the *Fundc1^-/-^* mice, the reductions in blood HCT and RBC, splenic erythroblast number and serum Epo concentration relative to WT mice are very mild. So while Fundc1 may have a role in maintaining Epo production during stress, the effect seems to be relatively minor, at least as revealed in the current assay. As the authors point out, Fundc1 may have a role in chronic stress erythropoiesis. In this regard, they may want to examine a more chronic model for anemia to examine the effects of modest Epo deficiency effects over time (see point #3).

2. To support further their hypothesis that delayed recovery from PHZ is not due to erythroid-intrinsic problems, the investigators should show that supplemental Epo administration eliminates the abnormalities observed in the *Fundc1^-/-^* mice.

3. The effects of Fundc1 loss on serum Epo levels after PHZ are relatively modest (Figure 2I), while the effects on kidney Epo mRNA are more extreme (Figure 3A). How can this discrepancy be explained? Perhaps measuring the kinetics of these parameters at a few timepoints after PHZ administration would help to reconcile these differences and be informative.

4. It might be useful to examine a different model for stress anemia. PHZ is a strong oxidant. Therefore, its effects on REPs could be due to oxidative stress rather than the erythropoietic demand of anemia. Investigating the effects of phlebotomy might be useful to distinguish the effects of oxidative stress from pure anemia. Additionally, repeated phlebotomy might allow the investigators to assess the role of Fundc1 in chronic anemia where the effects of Epo deficiency over time could have a more profound effect than what was observed after a single PHZ treatment.

5. The authors use one week of UUO treatment as a model for acute kidney disease. They show in this model that loss of Fundc1 causes enhanced accumulation of inflammatory markers, increased mitochondrial ROS and modest reductions in Epo compared to WT mice. However, they do not show that the model causes anemia in WT or Fundc1-/- mice. In order to substantiate a major claim of this manuscript, it is essential that the investigators examine the effects of Fund1c loss in a model for anemia of kidney disease, preferably a chronic one.

6. One central hypothesis is that Fundc1 deficiency promotes retention of abnormal mitochondria, inflammation and transformation to myofibroblast (abstract and line 470). However, the data do not show this. Rather, they report induction of some fibroblastic markers but constant numbers of REPs. Would it be possible to show this data?-- specifically co-staining of kidney cells for Epo mRNA or protein and PDGFb to quantify the number of REPs and their Epo induction.

7. In general, all of the histology slides in this manuscript are not well explained. For example, the histology in Figure S4C should be interpreted. What are the signs of renal injury? Is there evidence of injury throughout the kidney or specifically in Epo producing cells? The figure legends of all micrographs should include interpretation of the histopathology with the abnormal regions indicated on the micrograph.

8. Figure 3 Western blots – it is difficult to appreciate some of the differences in protein expression between WT vs mutant kidney. All western blots should include quantification of multiple experiments.

9. In several experiments, the authors interpret altered expression of p62 and LC3 as representing altered autophagy. This is not correct. Autophagy rates cannot be inferred by steady state measurements of these proteins. Enhanced autophagy could also deplete these proteins! Use of the Keima reporter seems more correct. Along the same lines, if Fundc1 is the mitochondrial autophagy adaptor in kidney, then what is the role of LC3 or p62 and how would their altered expression relate to mitophagy versus other autophagy pathways?

---

## [Author Response]

Summary:All the reviewers find the work of interest by analyzing the erythroid defects caused by deletion of mitophagy receptor FUNDC1 in mice. The study reveals that FUNDC1 loss results in the accumulation of impaired mitochondria in renal EPO producing cells, inflammation, myofibroblastic transformation, and reduced EPO production and anemia in experimental models of acute anemia or kidney injury. The results suggest that FUNDC1 is required for mitochondrial removal during erythropoiesis, and provide new insights into the anemia of kidney disease. The reviewers have raised several major concerns related to the strengths of the main conclusions. The complete comments are included.

We appreciate that the reviewers suggest our work provides “new insights into the anemia of kidney diseases”. Below, we have performed new experiments to address each of the reviewers’ comments and articulated the point-by-point responses.

Reviewer 1 also suggested that the title of the manuscript and other relevant texts need to be revised to better reflect the main findings or additional evidence is needed to support the conclusions related to stress erythropoiesis.

We agree with the reviewers and have changed the title from “Receptor-mediated mitophagy regulates stress erythropoiesis and confers protection against renal anemia” to “Receptor-mediated mitophagy regulates EPO production and protects against renal anemia”. We also generated the renal anemia (Figure 6) and chronic anemic mice model (Figure 2—figure supplement 1C-G) to further support our conclusions. In addition, the text was revised accordingly to reflect our findings.

Essential revisions:1. It is important to provide more convincing evidence to support the conclusions related to severe anemia or stress erythropoiesis observed in FUNDC1-deleted mice, or the major conclusions should be significantly revised.

Thanks for the insightful comment. To strengthen anemia observed in *Fundc1*^-/-^ mice upon stresses, we incorporated an additional renal anemia mouse model generated via the treatment with cisplatin (Figure 6). We indeed detected severer anemia in cisplatin*-Fundc1*^-/-^ mice than -WT controls (Figure 6A-C) (page 17, lines 436-442 (highlighted in blue)). We also revised the text to describe our results more precisely. For example, instead of stating “severe anemia”, we revised the text as “reduced RBC numbers and HCT values” in *Fundc1*^-/-^ mice upon PHZ-induced acute hemolytic anemia. (page 8, lines 156-159).

2. Evidence is needed to show whether defective mitophagy results in reduced Epo production, thus supporting the conclusion that the defective mitophagy observed in FUNDC1-deleted kidney is responsible for the decreased Epo production and associated phenotypes. A more direct experiment is required to assess the effect of PHZ on Epo production in Fundc1-/- kidney cells. To further support the hypothesis, supplemental Epo administration should be performed to examine the effects on the abnormalities observed in the Fundc1-/- mice.

We do appreciate these thoughtful suggestions. To directly assess the effects of PHZ on *Fundc1*^-/-^ renal cells, we FACS-sorted PDGFRb^+^ renal EPO producing cells (REPs) from kidneys of mito-Keima/WT and mito-Keima/*Fundc1*^-/-^ mice and treated them with PHZ, as described previously (1). We observed impaired mitophagy and reduced EPO production in *Fundc1*^-/-^ cells treated with a high dose of PHZ (100 mM), but not lower dose (10 mM) (Figure 4—figure supplement 1), suggesting that high dosage of PHZ might have, at least in part, a direct impact on the *Fundc1*^-/-^ REPs. The new data was incorporated in the revised manuscript on page 15, lines 376-385.

We also administrated EPO to PHZ-*Fundc1*^-/-^ mice (Figure 2J) and found that EPO supplement restored the RBC number, HGB and splenic erythroid differentiation to the similar level of WT controls (Figure 2J-L). These new data have been included in the revised manuscript on page 10-11, lines 238-244. We thus conclude that the defective mitophagy observed in *Fundc1*-deleted kidney is responsible for the decreased Epo production and associated phenotypes.

3. It is important to provide evidence for mitochondrial retention in a model for kidney disease that is accompanied by anemia. Such studies will strengthen the conclusion that FUNDC1 deficiency contributes to anemia associated with kidney disease by regulating mitophagy.

Thank you for the valuable comment. We utilized cisplatin in *Fundc1*^-/-^ mice to induce renal anemia (Figure 6). We found that cisplatin-treated *Fundc1*^-/-^ mice showed less RBCs and lower HGB in their peripheral blood (Figure 6B), which was accompanied by reduced serum EPO (Figure 6C). Also, we observed accumulated mitochondrial mass (Figure 6H), deficient mitophagy (Figure 6F-G), enhanced inflammation (Figure 6K) and decreased EPO production (Figure 6L) in PDGFRb^+^ REPs of cisplatin-treated mito-keima/*Fundc1^-/-^* mice. This new data was incorporated in the revised manuscript on page 17-18, lines 436-457.

4. Given the modest effects on blood cell production, the conclusions that FUNDC1-mediated mitophagy is not required for reticulocyte maturation need to be qualified or significantly revised.

The results from hemogram, including reduced RBCs and HGB as well as increased reticulocytes indicated that both PHZ-treated *Fundc1^-/-^* and WT mice were anemic (Figure 1—figure supplement 2A and Supplementary File 1), although little difference was found in RBCs, HGB and reticulocytes between them. We then isolated reticulocytes (CD71^+^Ter119^+^) in peripheral blood of PHZ-treated *Fundc1*^-/-^ and WT mice (Figure 1—figure supplement 2B) with morphologically confirmed by Giemsa and Brilliant Cresyl blue staining (Figure 1—figure supplement 2C), as previously described (2, 3). We cultured them to maturation *in vitro* for 3 days, using a previously described method (4). The brilliant cresyl blue staining showed progressive maturation of reticulocytes in WT mice after 3 days of culture (Figure 1—figure supplement 2C), while there were no detectable differences in the maturation of reticulocytes between WT and *Fundc1^-/-^* mice (Figure 1—figure supplement 2C). Consistent with these findings, mitochondrial mass showed little difference between WT and *Fundc1^-/-^* mice during reticulocyte maturation (Figure 1—figure supplement 2D). Thus, our results suggest that *Fundc1-*mediated mitophagy might be dispensable for reticulocyte maturation. The text for these data can be found on page 8, lines 163-169.

5. PHZ leads to significant reduction of Epo transcript in FUNDC1-deleted kidney cells but only a mild reduction of serum Epo level. This discrepancy should be explained.

We traced the kinetics of *Epo* mRNA and protein in kidney and serum, respectively, at multiple time points after PHZ treatment (Figure 2I, Figure 2—figure supplement 1A and Figure 3A). The data suggested that the discrepancy between EPO protein and mRNA likely arises from their asynchronization. Please see the text on page 10, lines 234-236.

6. Given the modest effects observed in FUNDC1-deleted mice and the limitations of the PHZ-induced acute anemia, it is important to include studies using a model for chronic anemia, and examine the effects of modest Epo deficiency effects over time. For example, phlebotomy may help to distinguish the effects of oxidative stress versus pure anemia. Repeated phlebotomy may help to assess the role of Fundc1 in chronic anemia could have a more profound effect than what was observed after a single PHZ treatment.

These comments are insightful. As suggested, we generated the chronic anemic model using phlebotomy. As shown in Figure 2—figure supplement 1C, we observed no significant differences in peripheral RBC counts, peripheral HGB level (Figure 2—figure supplement 1D), EPO production (Figure 2—figure supplement 1E) and erythroid differentiation in the spleen and BM (Figure 2—figure supplement 1F-G) between WT and *Fundc1*^-/-^ mice after phlebotomy. These results imply that *Fundc1* is inclined to respond to the oxidative stress of PHZ rather than pure anemia. Please see the text on page 11, lines 244-249.

7. Given the limitations of analyzing p62 and LC3 expression to represent altered autophagy, additional experiments are required to support conclusions related to autophagy.

We agree with the reviewers that the expression of P62 and LC3 are the biochemical hallmarks of general autophagy. To overcome this limitation, we have developed mt-Keima/*Fundc1*^-/-^ mice to examine mitophagy *in vivo* under various stress conditions (Figure 4H and 6F). We also measured the accumulation of mitochondrial membrane proteins (Figure 3E, 5D and 6D) by Western blotting. These evidences collectively reflect mitophagy in our system.

Reviewer #1:The authors have contributed substantially to the understanding of FUNDC1 as a mitophagic receptor in multiple cell types and disease conditions including hypoxia-mediated mitophagy in mammalian cells, inhibition of inflammasome activation, ischemia-reperfusion injury to the heart. Here the authors turned their attention to the kidney and showed that lack of FUNDC1 results in the accumulation of impaired mitochondria in EPO producing cells (REPs), leading to an inflammatory response, a myofibroblastic transformation of REPs, resulting in reduced EPO production and anemia in two experimental models including PHZ and UUO. Work presented here indicates that FUNDC1 does not play a role in mitochondrial removal during erythropoiesis. Overall, the paper is well written and presents strong evidence showing the importance of FUNDC1-mediated mitophage in renal fibrosis-mediated anemia by reducing the Epo production. However, the paper can be improvde by addressing following major concern:

We deeply appreciate the reviewer’s encouragement of our work and insightful comments.

The title and manuscript's emphasis of FUNDC1-mediaetd mitopahge regulting "stress eythropoiesis" are misleading. One of major complications of renal fibrosis is anemia due to the reduced production of Epo. Thus, simply because a particular protein deficiency results in renal fibrosis and reduced EPO production does not mean that such a protein should be considered as a regulator of "stress erythropoiesis" as implied by the title and multiple places in the manuscript. The findings showed here support that FUNDC1 protects against the accumulation of impaired mitochondria, inflammation, renal fibrosis and reduced EPO production and thus the development of major complication of anemia but not stress erythropoesis. Please consider to revise the title and text to avoid misleading and confusion.

We agree with the reviewer. We have accordingly revised the title as “Receptor-mediated mitophagy regulates EPO production and protects against renal anemia”. Please also refer to earlier comment for additional explanation.

Reviewer #2:The manuscript by Shi et al., analyzes the effect of deletion of mitophagy receptor FUNDC1 in mice on erythropoiesis and kidney production of erythropoietin (Epo). The authors have performed a considerable amount of work and generated a large set of data. Since the authors did not find any effect of FUNDC1 deletion on hematopoiesis or erythropoiesis, they subjected mice to erythroid stress by inducing hemolytic anemia using the established phenylhydrazine (PHZ) treatment. Mice survival in response to phenylhydrazine was reduced and attributed to severe anemia. They noticed a mild reduction in Epo production in response to phenylhydrazine. They then focused on kidney production of Epo. They confirm that FUNDC1-deleted kidney cells exhibit defective mitophagy, and PHZ treatment enhanced defective mitophagy in FUNDC1-deleted kidney cells. They attributed this defect to reduced Epo production.

We appreciate the positive comments on our manuscript.

There are two major problems in this manuscript. One is that in contrary to the authors assertion, it is unclear whether FUNDC1-deleted mice exhibit anemia (hemoglobin is not reported in PHZ-treated mice and other RBC indices are very mild) but possibly only a reduced compensated red blood cell production in response to phenylhydrazine treatment.

Thanks for the thoughtful comments. We have provided the hemogram of PHZ-treated *Fundc1*^-/-^ mice, indicating that these mice were anemic (Supplementary File 1 and Figure 1—figure supplement 2A). To further strengthen the anemia observed in the *Fundc1*^-/-^ mice upon stresses, we treated *Fundc1*^-/-^ mice with cisplatin, which induced the renal anemia. We indeed detected severer anemia in *Fundc1*^-/-^ mice than WT controls (please refer to Essential revision 2 above). However, to describe our results more precisely, we also revised the text as “reduced RBC number and HCT values” in *Fundc1*^-/-^ mice upon PHZ-induced acute hemolytic anemia.

The other is that although the authors confirm that FUNDC1-deleted kidney cells exhibit defective mitophagy that is more pronounced in PHZ-treated animals, they do not show that reduced Epo production results from defective mitophagy. My specific comments are below:

We thank the reviewer for pointing it out. The EPO expression was reduced in the PDGFRb^+^ REPs of PHZ- (Figure 4M), UUO- (Figure 5I) and cisplatin-treated (Figure 6L) *Fundc1*^-/-^ mice, respectively. Meanwhile, all these PDGFRb^+^ REPs exhibited defective mitophagy upon PHZ- (Figure 4H,I), UUO- (Figure 5D) and cisplatin-treated (Figure 6F,G). In addition, the direct effect of PHZ on the EPO production was also examined (Figure 4—figure supplement 1). Please refer to Essential revision 2 above.

– There is no evidence of mitochondrial retention in the spleen, BM and PB. The authors use CD71 as a marker but it is unclear on which cells.

We replaced CD71 with Ter119 to better distinguish erythroid cells (Figure 1—figure supplement 1B-D). We found that there were no evident changes between *Fundc1*^-/-^ mice and WT control despite substantial mitochondrial retention exhibited in *Bnip3l*^-/-^ mice (Figure 1—figure supplement 1B-D).

– Although the survival in response to phenylhydrazine is reduced which was achieved by using 100mg/kg (instead of the usual dose of 50mg/kg) and including 30 mice (p=0.04), there was no severe anemia. In fact Hb levels are normal and the slight reduction in red blood cell indices suggest that there was a mild reduced red blood cell production that did not reach the definition of anemia. So, in the absence of anemia the claim that Fundc1-mediated mitophagy is not required for reticulocyte maturation is not accurate.

Please refer to Essential revision 4.

– It remains unclear what animals die from.

FUNDC1 is expressed in a broad range of tissues and organs, playing an important role in cardiac tissues, liver and kidney (8-10). Currently, the cause leading to the death of the animal is yet to be determined. We speculate that the death of PHZ-*Fundc1*^-/-^ mice might result from the dysfunction of multiple tissues and/or organs beyond anemia.

– The authors FACS-sort reticulocytes 5 days post phenylhydrazine. This is an interesting experiment. What conditions for the FACS sorting do they use?

Please refer to Essential revision 4.

– From Figure S2 it seems that the authors FACS-sorted immature erythroblasts but not reticulocytes, the data on mitochondria in these cells also seems to be from immature erythroid cells. The authors should clarify how they distinguish reticulocytes from immature erythroblasts.

Please refer to Essential revision 4.

– The authors state: "Together, our results suggest that FUNDC1- mediated mitophagy isdispensable for programmed mitochondrial elimination in reticulocytes duringerythroid maturation." It is difficult to assess the validity of this statement given that it is unclear which cells were used in these experiment.

Please refer to Essential revision 4.

– The authors examine differential gene expression in R2 erythroblast subpopulation between cells from mice subjected to PHZ treatment or not. They use RNA-Seq analysis of R2 and identify among others JAK-STAT pathway to be differentially expressed in PHZ treated Fundc1-/- spleens. They conclude that JAK-STAT is hypoactivated in Fundc1-/- R2 cells. Given that the reduced RBC production seem to be compensated (see R3, R4) does not seem to be highly affected this result seems to indicate compensated JAK-STAT hypoactivation.

The R3 and R4 populations appears to be normal in the spleen of PHZ-*Fundc1*^-/-^ and -WT erythroid cells, suggesting that other signaling pathways may compensate the hypoactivation of the JAK-STAT signaling pathway. However, since we have not performed the RNA-Seq analysis of R3 and R4 cells from *Fundc1^-/-^* and WT mice, we do not know yet what signal(s) may compensate the hypoactivated JAK-STAT signaling pathway. This topic should be addressed in future investigations.

– How do the authors explain that Epo reduction in Fundc1-/- cells has an effect solely on R2 spleen cells?

It has been reported that EPOR, a transmembrane glycoprotein, is expressed in erythroid progenitors (colony-forming unit-erythroid: CFU-E), proerythroblasts, and early basophilic erythroblasts (5, 6). The proerythroblasts and early basophilic erythroblasts are the major components of R2 population, as indicated by Giemsa staining of sorted R2 cells (Author response image 1) as well as prior studies (7). Given that the major components of R2 are EPO dependent, which may account for the sensitivity of R2 cells to EPO paucity. Please see the text on page 10, lines 224-229.

**Author response image 1. sa2fig1:** Giemsa staining of FACS-sorted R2 populations. The red and green arrow indicate proerythroblasts and basophilic erythroblasts, respectively. The photographs were taken at an original magnification of 400×.

– The significant reduction of Epo transcript in kidney cells in response to PHZ is interesting but how does this correlate with a mild reduction of Epo in PB?

Please refer to Essential revision 5.

– To determine the role of defective mitophagy on Epo production, the authors selected PDGFRB+ and – cells and subjected to hypoxia. A more direct experiment is to assess the effect of PHZ on Epo production in Fundc1-/- kidney cells instead they have measured mitochondrial mass which does not address their hypothesis.

Please refer to Essential revision 2.

– How are experiments in 4M and 3A different? What is missing is the effect of PHZ on Fundc1-/- PDGFRB+ cells?

In Figure 3A the *Epo* level was tested with the lysates of whole kidney, while in Figure 4M the *Epo* level was measured in FACS-sorted PDGFRb^+^ REPs.

Please refer to Essential revision 2 for the direct effect of PHZ on *Fundc1^-/-^* PDGFRb^+^ cells.

Reviewer #3:The authors show that loss of the mitophagy receptor Fundc1 in mice causes defects in stress erythropoiesis by impairing the production of Epo by the kidney, with defective mitophagy of Epo-producing cells. This is interesting work that has the potential to provide insights into the anemia of kidney disease. However, the overall effects seem to be fairly modest and several of the studies must be investigated further to support the claims to a reasonable level. Notably, modest impairment of Epo production could lead to significant anemia over time, but this is not shown. Perhaps the most important problem is that study does not examine a model for anemia of kidney disease.

We thank the reviewer for the insightful comments and suggestions.

1. Following PHZ administration to the Fundc1-/- mice, the reductions in blood HCT and RBC, splenic erythroblast number and serum Epo concentration relative to WT mice are very mild. So while Fundc1 may have a role in maintaining Epo production during stress, the effect seems to be relatively minor, at least as revealed in the current assay. As the authors point out, Fundc1 may have a role in chronic stress erythropoiesis. In this regard, they may want to examine a more chronic model for anemia to examine the effects of modest Epo deficiency effects over time (see point #3).

Please refer to Essential revision 6.

2. To support further their hypothesis that delayed recovery from PHZ is not due to erythroid-intrinsic problems, the investigators should show that supplemental Epo administration eliminates the abnormalities observed in the Fundc1-/- mice.

Please refer to Essential revision 2.

3. The effects of Fundc1 loss on serum Epo levels after PHZ are relatively modest (Figure 2I), while the effects on kidney Epo mRNA are more extreme (Figure 3A). How can this discrepancy be explained? Perhaps measuring the kinetics of these parameters at a few timepoints after PHZ administration would help to reconcile these differences and be informative.

Please refer to Essential revisions.

4. It might be useful to examine a different model for stress anemia. PHZ is a strong oxidant. Therefore, its effects on REPs could be due to oxidative stress rather than the erythropoietic demand of anemia. Investigating the effects of phlebotomy might be useful to distinguish the effects of oxidative stress from pure anemia. Additionally, repeated phlebotomy might allow the investigators to assess the role of Fundc1 in chronic anemia where the effects of Epo deficiency over time could have a more profound effect than what was observed after a single PHZ treatment.

Please refer to Essential revision 6.

5. The authors use one week of UUO treatment as a model for acute kidney disease. They show in this model that loss of Fundc1 causes enhanced accumulation of inflammatory markers, increased mitochondrial ROS and modest reductions in Epo compared to WT mice. However, they do not show that the model causes anemia in WT or Fundc1-/- mice. In order to substantiate a major claim of this manuscript, it is essential that the investigators examine the effects of Fund1c loss in a model for anemia of kidney disease, preferably a chronic one.

We have generated a cisplatin-induced renal anemic model to measure the role of Fundc1. Please refer to Essential revision 1 and Essential revision 3.

6. One central hypothesis is that Fundc1 deficiency promotes retention of abnormal mitochondria, inflammation and transformation to myofibroblast (abstract and line 470). However, the data do not show this. Rather, they report induction of some fibroblastic markers but constant numbers of REPs. Would it be possible to show this data?-- specifically co-staining of kidney cells for Epo mRNA or protein and PDGFb to quantify the number of REPs and their Epo induction.

It is an excellent idea. We found the percentage of REPs show no difference between WT and *Fundc1*^-/-^ mice after PHZ treatment in kidney as Author response image 2, despite the capacity of EPO production was reduced (Figure 4M, 5I and 6L). As suggested, we had tried the co-staining of Epo mRNA and PDGFRb protein in FACS-sorted PDGFRb^+^ REP cells in PHZ-*Fundc1*^-/-^ and -WT mice. However, the EPO RNA-FISH experiments did not work out. We speculate that this might result from the lengthy procedure for renal sample preparation, which led to the degradation of EPO mRNA.

**Author response image 2. sa2fig2:** The percentage of REPs after PHZ treatment in kidney.

7. In general, all of the histology slides in this manuscript are not well explained. For example, the histology in Figure S4C should be interpreted. What are the signs of renal injury? Is there evidence of injury throughout the kidney or specifically in Epo producing cells? The figure legends of all micrographs should include interpretation of the histopathology with the abnormal regions indicated on the micrograph.

The interpretation of histopathology was incorporated in the figure legends (Figure 3—figure supplement 1C) of the revised manuscript. (Page 44, lines 1208-1215).

8. Figure 3 Western blots – it is difficult to appreciate some of the differences in protein expression between WT vs mutant kidney. All western blots should include quantification of multiple experiments.

We have included the quantification of multiple experiments in all Western blots.

9. In several experiments, the authors interpret altered expression of p62 and LC3 as representing altered autophagy. This is not correct. Autophagy rates cannot be inferred by steady state measurements of these proteins. Enhanced autophagy could also deplete these proteins! Use of the Keima reporter seems more correct. Along the same lines, if Fundc1 is the mitochondrial autophagy adaptor in kidney, then what is the role of LC3 or p62 and how would their altered expression relate to mitophagy versus other autophagy pathways?

Please refer to Essential revision 7.

With respect to “if Fundc1 is the mitochondrial autophagy adaptor in kidney, then what is the role of LC3 or P62”, the expression of P62 and LC3 is the biochemical hallmark of general autophagy and selective mitophagy also needs general autophagy machinery. For example, our prior studies demonstrated that P62 (15) and LC3 (16) are involved in FUNDC1-mediated mitophagy.

With respect to “how would their altered expression relate to mitophagy versus other autophagy pathways?”, although we cannot completely rule out the possibility that P62 and LC3 might also be involved in other autophagic pathways, the collective results from mt-Keima/*Fundc1*^-/-^ mice (Figure 4H and 6F) and the accumulation of mitochondrial membrane proteins (Figure 3E, 5D and 6D) suggest that the altered expression of P62 and LC3 could partially reflect deficient mitophagy with Fundc1 deletion.

1. Merle NS, Grunenwald A, Figueres ML, Chauvet S, Daugan M, Knockaert S, et al. Characterization of Renal Injury and Inflammation in an Experimental Model of Intravascular Hemolysis. *Front Immunol.* 2018;9:179.2. Sandoval H, Thiagarajan P, Dasgupta SK, Schumacher A, Prchal JT, Chen M, et al. Essential role for Nix in autophagic maturation of erythroid cells. *Nature.* 2008;454(7201):232-5.3. Gothwal M, Wehrle J, Aumann K, Zimmermann V, Grunder A, and Pahl HL. A novel role for nuclear factor-erythroid 2 in erythroid maturation by modulation of mitochondrial autophagy. *Haematologica.* 2016;101(9):1054-64.4. Koury MJ, Koury ST, Kopsombut P, and Bondurant MC. In vitro maturation of nascent reticulocytes to erythrocytes. *Blood.* 2005;105(5):2168-74.5. Wickrema A, Bondurant MC, and Krantz SB. Abundance and stability of erythropoietin receptor mRNA in mouse erythroid progenitor cells. *Blood.* 1991;78(9):2269-75.6. Broudy VC, Lin N, Brice M, Nakamoto B, and Papayannopoulou T. Erythropoietin receptor characteristics on primary human erythroid cells. *Blood.* 1991;77(12):2583-90.7. Socolovsky M, Nam H, Fleming MD, Haase VH, Brugnara C, and Lodish HF. Ineffective erythropoiesis in Stat5a(-/-)5b(-/-) mice due to decreased survival of early erythroblasts. *Blood.* 2001;98(12):3261-73.8. Wu S, Lu Q, Wang Q, Ding Y, Ma Z, Mao X, et al. Binding of FUN14 Domain Containing 1 With Inositol 1,4,5-Trisphosphate Receptor in Mitochondria-Associated Endoplasmic Reticulum Membranes Maintains Mitochondrial Dynamics and Function in Hearts in Vivo. *Circulation.* 2017;136(23):2248-66.9. Li W, Li Y, Siraj S, Jin H, Fan Y, Yang X, et al. FUN14 Domain-Containing 1-Mediated Mitophagy Suppresses Hepatocarcinogenesis by Inhibition of Inflammasome Activation in Mice. *Hepatology.* 2019;69(2):604-21.10. Wang J, Zhu P, Li R, Ren J, and Zhou H. Fundc1-dependent mitophagy is obligatory to ischemic preconditioning-conferred renoprotection in ischemic AKI via suppression of Drp1-mediated mitochondrial fission. *Redox Biol.* 2020;30:101415.11. Greenwald AC, Licht T, Kumar S, Oladipupo SS, Iyer S, Grunewald M, et al. VEGF expands erythropoiesis via hypoxia-independent induction of erythropoietin in noncanonical perivascular stromal cells. *J Exp Med.* 2019;216(1):215-30.12. Broeker KAE, Fuchs MAA, Schrankl J, Kurt B, Nolan KA, Wenger RH, et al. Different subpopulations of kidney interstitial cells produce erythropoietin and factors supporting tissue oxygenation in response to hypoxia in vivo. *Kidney Int.* 2020;98(4):918-31.13. Gerl K, Nolan KA, Karger C, Fuchs M, Wenger RH, Stolt CC, et al. Erythropoietin production by PDGFR-beta(+) cells. *Pflugers Arch.* 2016;468(8):1479-87.14. Pan X, Suzuki N, Hirano I, Yamazaki S, Minegishi N, and Yamamoto M. Isolation and characterization of renal erythropoietin-producing cells from genetically produced anemia mice. *PLoS One.* 2011;6(10):e25839.15. Li Y, Xue Y, Xu X, Wang G, Liu Y, Wu H, et al. A mitochondrial FUNDC1/HSC70 interaction organizes the proteostatic stress response at the risk of cell morbidity. *EMBO J.* 2019;38(3).16. Liu L, Feng D, Chen G, Chen M, Zheng Q, Song P, et al. Mitochondrial outer-membrane protein FUNDC1 mediates hypoxia-induced mitophagy in mammalian cells. *Nat Cell Biol.* 2012;14(2):177-85.17. Chang YT, Yang CC, Pan SY, Chou YH, Chang FC, Lai CF, et al. DNA methyltransferase inhibition restores erythropoietin production in fibrotic murine kidneys. *J Clin Invest.* 2016;126(2):721-31.18. Schweers RL, Zhang J, Randall MS, Loyd MR, Li W, Dorsey FC, et al. NIX is required for programmed mitochondrial clearance during reticulocyte maturation. *Proceedings of the National Academy of Sciences of the United States of America.* 2007;104(49):19500-5.